# Eleven New Species of the Genus *Tarzetta* (Tarzettaceae, Pezizales) from Mexico

**DOI:** 10.3390/jof10060403

**Published:** 2024-06-04

**Authors:** Marcos Sánchez-Flores, Jesús García-Jiménez, Tania Raymundo, César R. Martínez-González, Juan F. Hernández-Del Valle, Marco A. Hernández-Muñoz, Javier I. de la Fuente, Martín Esqueda, Alejandrina Ávila Ortiz, Ricardo Valenzuela

**Affiliations:** 1Herbario Micológico José Castillo Tovar, Instituto Tecnológico de Ciudad Victoria, Tecnológico Nacional de México, Boulevard Emilio Portes Gil No. 1301, Ciudad Victoria 87010, Tamaulipas, Mexico; sanflores37@gmail.com (M.S.-F.); cesar.ramiro.mg@gmail.com (C.R.M.-G.); sanfrancisco_hv89@hotmail.com (J.F.H.-D.V.); 2Laboratorio de Micología, Escuela Nacional de Ciencias Biológicas, Instituto Politécnico Nacional, Prolongación de Carpio and Plan de Ayala, Santo Tomás, Alcaldía Miguel Hidalgo, Ciudad de Mexico 11340, Mexico; traymundoo@ipn.mx (T.R.); rvalenzg@ipn.mx (R.V.); 3Herbario FEZA, Facultad de Estudios Superiores Zaragoza, Universidad Nacional Autónoma de México, Batalla de 5 de Mayo s/n, Colonia Ejercito de Oriente, Alcaldía Iztapalapa, Ciudad de Mexico 09230, Mexico; marcohm13@yahoo.com.mx (M.A.H.-M.); aviort27@gmail.com (A.Á.O.); 4Edafología, Campus Montecillo, Colegio de Postgraduados, Km 36.5, Montecillo, Texcoco 56230, Estado de México, Mexico; jdelafuenteitcv@gmail.com; 5Centro de Investigación en Alimentación y Desarrollo A.C., Carretera Gustavo Enrique Astiazarán Rosas 46, La Victoria, Hermosillo 83304, Sonora, Mexico; esqueda@ciad.mx

**Keywords:** ectomycorrhizal fungi, phylogeny, taxonomy, temperate forest

## Abstract

The genus *Tarzetta* is distributed mainly in temperate forests and establishes ectomycorrhizal associations with angiosperms and gymnosperms. Studies on this genus are scarce in México. A visual, morphological, and molecular (ITS-LSU) description of *T. americupularis*, *T. cupressicola*, *T. davidii*, *T. durangensis*, *T. mesophila*, *T. mexicana*, *T. miquihuanensis*, *T. poblana*, *T. pseudobronca*, *T. texcocana*, and *T. victoriana* was carried out in this work, associated with *Abies*, *Quercus*, and *Pinus*. The results of SEM showed an ornamented ascospores formation by Mexican Taxa; furthermore, the results showed that *T. catinus* and *T. cupularis* are only distributed in Europe and are not associated with any American host.

## 1. Introduction

Recently, the family Tarzettaceae (Pezizales, Pezizomycetes) was erected by Ekanayaka et al. [1] based on multigene phylogenetic analysis (ITS, LSU, SSU, and *tef1-α*, *rpb2*) and was segregated from the family Pyronemataceae according to Perry et al. [2]. Previously, the family Pyronemataceae was considered polyphyletic by Hansen et al. [3] with the *Geopyxis* lineage and by Kumar et al. [4] with the *Tarzetta–Geopyxis* lineage. Currently, this family is represented by *Tarzetta* (Cooke) Lambotte as the type genus *Geopyxis* (Pers.) Sacc, *Hydnocystis* Tul. & C. Tull., *Hypotarzetta* Donadini, *Paurocotylis* Berk., and *Stephensia* Tull. & C. Tul. [1]. However, the Index Fungorum (https://www.indexfungorum.org/, accessed on 15 January 2024) still considers it a monogeneric family. The genus *Tarzetta* has a restricted distribution mostly in temperate forests, which forms ectomycorrhizal associations, generally with trees and shrubs of the genera *Abies* Mill., *Alnus* Mill., *Quercus* L., *Pinus* L., and *Pseudotsuga* Carrière [3,5,6,7]. *Tarzetta* species are characterized as small to medium apothecia (2–30 mm), sessile to stipitate, deeply cupulate, and grey to beige but sometimes ochraceous or yellowish and rarely orange. Most of the species present a hymenium whitish or concolorous to the external zone of the apothecia, with a margin that is usually denticulate. Microscopically, it has simple to branched paraphyses, septate, and hyaline; asci operculated, inamyloid, 8-spored. Ascospores are broadly ellipsoid, ellipsoid to oblong-ellipsoid, single to bigutulated, and usually smooth although *T. jafneospora* W.Y. Zhuang & Jorf. has verrucose ornamentation [3,7].

Twenty-three species of *Tarzetta* have been described worldwide: 16 species from Europe: *T. alnicola* Van Vooren, *T. alpina* Van Vooren & Cheype, *T. catinus* (Holmsk.) Korf & J.K. Rogers, *T. cupularis* (L.) Lambotte, *T. gaillardiana* (Boud.) Korf & J.K. Rogers, *T. gregaria* Van Vooren, *T. melitensis* Sammut & Van Vooren, *T. oblongispora* M. Carbone, S. Saitta, L. Sánchez, García Blanco & Van Vooren, *T. ochracea* (Gillet) Van Vooren, *T. pusilla* Harmaja, *T. quercus-ilicis* Van Vooren & M. Carbone, *T. scotica* (Rea) Y.J. Yao & Spooner, *T. sepultarioides* Van Vooren, *T. spurcata* (Pers.) Harmaja, and *T. velata* (Quél.) Svrček; 4 species from China: *T. confusa* F.M. Yu, S. Wang, Q. Zhao & K.D. Hyde, *T. linzhiensis* F.M. Yu, S. Wang, Q. Zhao & K.D. Hyde, *T. tibetensis* F.M. Yu & Q. Zhao, and *T. urceolata* L. Lei & Q. Zhao; 3 species from the Americas: *T. brasiliensis* Rick from Brasil, *T. microspora* (Raithelh.) Raiithelh, from Argentina, and *T. bronca* (Peck) Korf & J.K. Rogers from the USA; and 1 species from New Zealand: *T. jafneospora* [7,8,9,10,11].

Studies on *Tarzetta* are scarce in Mexico. Some collections of *T. catinus* and *T. cupularis* are known from the states of Durango, Hidalgo, Mexico City, Mexico State, Michoacán, Morelos, Oaxaca, and Veracruz [12,13,14,15,16,17,18,19]; nevertheless, those collections lack complete taxonomic descriptions. Through molecular studies, Van Vooren et al. [7] sorted *T. catinus* and *T. cupularis* from the species previously cited with these names for the Americas, mentioning that the taxa in the continent should have greater taxonomic and phylogenetic exploration. This study aims to describe 11 new species of *Tarzetta* from Mexico, using morphological and molecular data.

## 2. Materials and Methods

### 2.1. Colecting and Morphology

The specimens were collected in temperate forests from 2019 to 2023. The collected specimens were deposited in the José Castillo Tovar mycological herbarium of the Insti-tuto Tecnológico de Ciudad Victoria (ITCV), the herbarium of the Escuela Nacional de Ciencias Biológicas of the Instituto Politécnico Nacional (ENCB), the herbarium of the Facultad de Estudios Superiores Zaragoza (FEZA) of the Universidad Nacional Autónoma de México (UNAM), and the Mycological Collection of the Universidad Estatal de Sonora (UES). Further, herbarium specimens were analyzed in ENCB, ITCV, and FEZA Herbaria.

The macroscopic morphological characteristics of specimens such as size, shape, and color were described [7]. The Illustrated Dictionary of Mycology was used for morphological terminology [20]. Apothecia colors are described according to Kornerup and Wanscher [21]. Longitudinal cuts of the apothecia were made and rehydrated with 70% alcohol, 5% KOH, water, and cotton blue to observe a possible ornamentation of the ascospores. The microscopic characters such as excipulum, paraphyses, asci, and ascospores were characterized for identification using an optical microscope (OM) (Axiostar plus, Zeiss, Jena, Germany; VE-B1, Velab, Ciudad de México, Mexico). The photographs were taken with a Rebel T-1i camera, a 100 mm macro lens (Canon, Tokyo, Japan), and a DCS-W630 camera (SONY, Tokyo, Japan). Scanning Electron Microscopy (SEM; SU1510, Hitachi High Technologies, Tokyo, Japan) was used to observe the ornamentation of the ascospores.

### 2.2. Extraction, Amplification, and Sequencing of DNA

The DNA was extracted from dried herbarium specimens. Genomic DNA was extracted using the CTAB method [22]. Two molecular markers were used, and these were the Internal Transcribed Spacer region of nuclear ribosomal DNA (ITS1-5.8-ITS2 nrDNA; hereafter ITS) and the large subunit nrDNA (28S). PCR amplification included 35 cycles with an annealing temperature of 54 °C. It was carried out with the ITS5 and ITS4 primers [23] for the ITS nrDNA region and the LROR and LR5 primers [24] for the 28S nrDNA region (LSU). The PCR products were verified using agarose gel electrophoresis. The gels were run for 1 h at 95 V cm^−3^ in 1.5% agarose and 1× TAE buffer (Tris Acetate-EDTA, Saint Lois, MO, USA). The gel was stained with GelRed (Biotium, Fremont, CA, USA) and the bands were visualized in an Infinity 3000 transilluminator (Vilber Lourmat, Baden-Wurtemberg, Germany). The amplified products were purified with the ExoSAP Purification kit (Affymetrix, Santa Clara, CA, USA), following the manufacturer’s instructions. They were quantified and prepared for the sequence reaction using a BigDye Terminator v. 3.1 (Applied Biosystems, Waltham, MA, USA). These products were sequenced in both directions with an Applied Biosystems model 3730XL (Applied BioSystems, Waltham, MA, USA) at the Instituto de Biología of the Universidad Nacional Autónoma de México (UNAM).

### 2.3. Sequence Assembly

The sequences of both strands of each of the genes were analyzed, edited, and assembled using BioEdit version 7.0.5 [25] to generate a consensus sequence, which was compared with those deposited in the GenBank of the National Center for Biotechnology Information (NCBI), using the tool BLASTN 2.2.19 [26].

### 2.4. Phylogenetic Analysis

To study phylogenetic relationships, our newly produced sequences of twenty-six individuals were added to reference sequences of ITS and LSU nrDNA deposited in the NCBI database (http://www.ncbi.nlm.nih.gov/genbank/, accessed on 25 January 2024), and an alignment was performed based on the taxonomic sampling employed by Van Vooren et al. [7] and Healy et al. [27] (Table 1). Each region was aligned using the online version of MAFFT v. 7 [28,29]. The alignment was revised in PhyDE v. 10.0 [30], followed by minor manual adjustments to ensure character homology between taxa. The matrix was composed for ITS by 55 taxa (692 characters) and LSU by 61 taxa (800 characters). The data were analyzed using maximum parsimony (MP), maximum likelihood (ML), and Bayesian inference (BI). Maximum parsimony analyses were carried out in PAUP* 4.0b10 [31] using the heuristic search mode, 1000 random starting replicates, and TBR branch swapping with MULTREES and Collapse on.

Bootstrap values were estimated using 1000 bootstrap replicates under the heuristic search mode, each with 100 random starting replicates. Maximum likelihood analyses were carried out in RAxML v. 8.2.10 [32] with a GTR + G model of nucleotide substitution. To assess branch support, 10,000 rapid bootstrap replicates were run with the GTRGAMMA model. Bayesian inference was carried out in MrBayes v. 3.2.6 x64 [33] with four chains, and the best evolutionary model for alignment was sought using PartitionFinder v. 2 [34,35,36]. The information block for the matrix includes two simultaneous runs, four Montecarlo chains, a temperature set to 0.2, and a sampling of 10 million generations (standard deviation ≤ 0.1) with trees sampled every 1000 generations. The first 25% of samples were discarded as burn-in, and convergence was evaluated by examining the standard deviation of split frequencies among runs and by plotting the log-likelihood values from each run using Tracer v. 1 [37]. The remaining trees were used to calculate a 50% majority-rule consensus topology and posterior probabilities (PP). Trees were visualized and optimized in FigTree v. 1.4.4 [38].

## 3. Results

### 3.1. Molecular Analysis

Phylogenetic reconstruction was based on the alignment of the nrITS + LSU dataset (56 taxa, 1520 characters, including gaps). The three phylogenetic analyses of the dataset, MP, ML, and BI, recovered similar topologies (Figure 1). No significant conflict (bootstrap value > 70%) was detected among the topologies obtained via separate phylogenetic analyses. The parsimony analysis of the alignment found 1205 trees of 291 steps (CI = 0.5022, HI = 0.1475, RI = 0.4785, RC = 0.3785). The best RA×ML tree with a final likelihood value of –44,572.924927 is presented. The matrix had 1095 distinct alignment patterns, with 5.15% undetermined characters or gaps. Estimated base frequencies were as follows: A = 0.114712, C = 0.191626, G = 0.180634, T = 0.213028; substitution rates AC = 1.007806, AG= 1.154719, AT = 1.290447, CG = 1.045887, CT = 4.696475, GT = 1.000000; and gamma distribution shape parameter α= 0.002898. In the Bayesian analysis, the standard deviation between the chains stabilized at 0.00002 after 3 million generations. No significant changes in tree topology trace or cumulative split frequencies of selected nodes were observed after approximately 0.25 million generations, which were discarded as 25% burn-in.
Figure 1Maximum likelihood phylogeny based on the nrITS + LSU sequence data. Maximum parsimony and Bayesian analyses recovered identical topologies concerning the relationships among the main clades of the *Tarzetta*. For each node, the following values are provided: maximum parsimony bootstrap (%)/maximum likelihood bootstrap (%)/ and posterior confidence (*p*-value). The scale bar represents the expected number of nucleotide substitutions per site. The new species of *Tarzetta* are shown in bold.
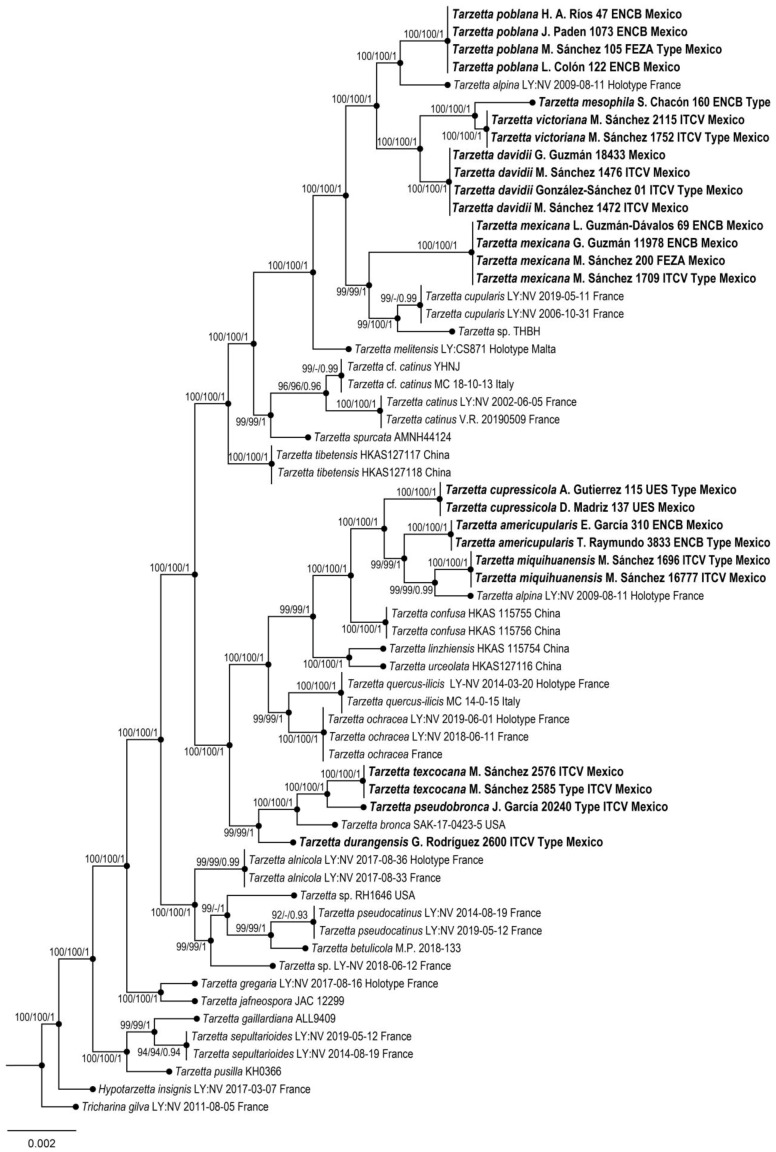



### 3.2. Taxonomy

Eleven species of the genus *Tarzetta* are described as a new species, and they are based on morphological, ecological, and molecular characteristics. Furthermore, a map of Mexico shows the distribution of the type species (Figure 2); a comparative table of Mexican species and some American and European species with morphological and ecological characteristics (vegetation type and ectomycorrizal host) (Table 2) and the taxonomic key of the Mexican species of *Tarzetta* are included.
Figure 2Localities of the type species in Mexico.
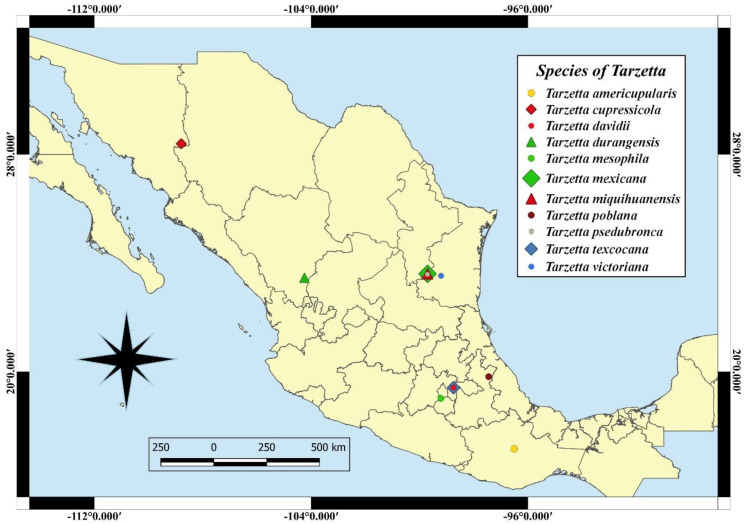



*Tarzetta americupularis* Sánchez-Flores, García-Jiménez, R. Valenz. & Raymundo, sp. nov. (Figure 3, Figure 14A and Figure 15A)

Mycobank: #851432

Diagnosis: Apothecia 4–8 mm diameter, hymenium pale orange, margin crenate, granulate to prunoise; ascospores (15–) 17–25 (–26) × 9–13 (–14) µm, ellipsoid to oblong, grown under *Quercus* spp.

Type: MEXICO: Oaxaca state. Santa Catarina Lachatao municipality, Santa Martha Latuvi, place 3 caminos, La Muralla (17°09′43″ N, 96°30′35.4″ W), 2700 m asl, 2 September 2011, T. Raymundo 3833 (ENCB, holotype).

GenBank: ITS: PP825384, LSU: PP825427.

Etymology: The epithet refers to the morphological similarity with *T. cupularis* (L.) Lambotte but occurs on the American Continent.
Figure 3*Tarzetta americupularis*. (**A**) Apothecium; (**B**) longitudinal section of the apothecium; (**C**) hymenium; (**D**) ectal excipulum cells; (**E**) asci and ascospores; and (**F**) ascospores.
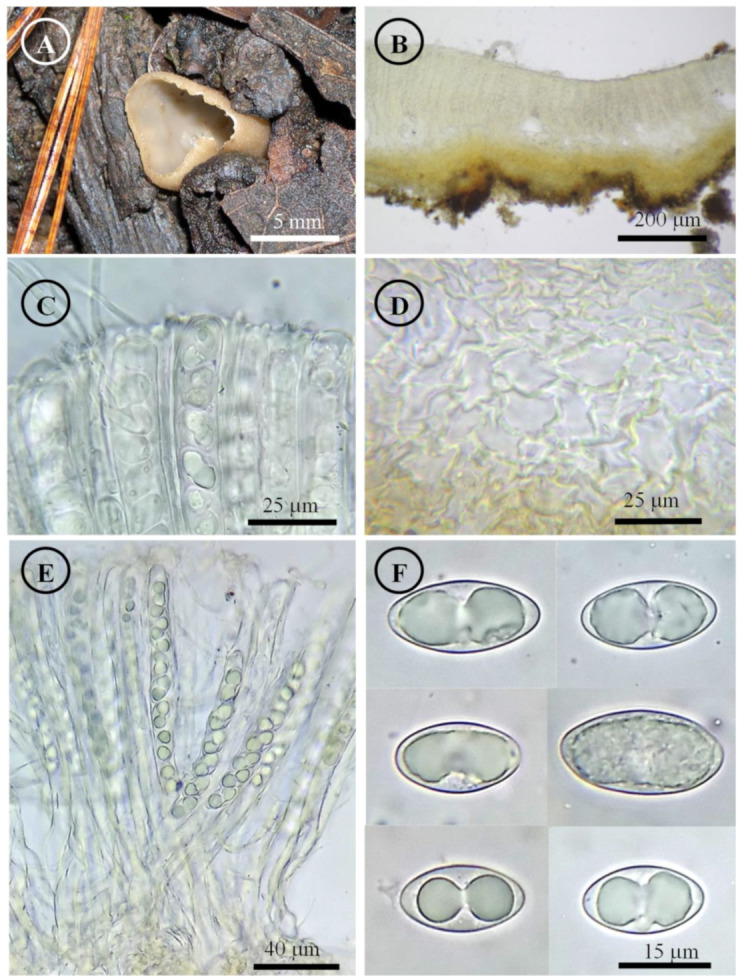



Apothecia 4–8 mm in diameter, cupuliform, solitary to scattered, sessile, hymenium pale orange (5A3), margin crenate that tears at maturity, external surface greyish orange (5B4), granulate to pruinose, asperulate. Ectal excipulum 30–90 µm thick, *textura angularis* with cells 12–27 × 9–18 µm, hyaline, thin-walled. Medullary excipulum 35–90 µm thick, *textura intricata* with hyphae 3–6 µm in diameter, hyaline. Subhymenium 8–20 µm thick. Hymenium 200–315 µm thick. Paraphyses 2–4 µm in diameter, filiform, septate, with rounded to abrupt apex. Asci 190–310 (–325) × 11–15 µm, cylindrical, 8-spored, hyaline, inamyloid. Ascospores (15–) 17–25 (–26) × 9–13 (–14) µm [x = 20.5 × 11.5 µm, *n* = 63], Q = (1.5–) 1.6–2.2 (–2.3) Qm = 1.9, ellipsoid to oblong, hyaline, 1–2 guttules, smooth on OM, very finely rugose on SEM.

Habit: On soil, in *Pinus-Quercus* and coniferous forest.

Distribution: MEXICO. Mexico City, Mexico State, and Oaxaca.

Material examined: Mexico, Mexico City, Álvaro Obregón town hall, Desierto de los Leones, 1 September 1972, E. García 310 (ENCB, paratype; ITS: PP825385, LSU: PP825428). Mexico State, Donato Guerra municipality, El Capulín, 25 September 1983, R.E. Santillán 487 (ENCB). Huixquilucan municipality, Santa Cruz, 23 September 2023, T. Raymundo 9367 (ENCB). Jilotepec municipality, El Cuzda hill, México-Querétaro highway, 28 August 1983, L. Guzmán-Dávalos 753 (ENCB). Tlalmanalco municipality, road Tlalmanalco-San Rafael, 25 September 1955, G. Guzmán 336-D (ENCB). Oaxaca state, Santa Catarina Ixtepeji municipality, Centro Ecoturistico La Cumbre, 7 September 2023, M. Sánchez 3278 (ITCV), 3281 (ITCV), 3285 (ITCV), 3288 (ITCV), 3294 (ITCV).

Notes: This species is characterized by forming apothecia 4–8 mm in diameter and ellipsoid to oblong ascospores [Q = (1.5–) 1.6–2.2 (–2.3) µm]. Microscopically, it is similar to *T. cupularis*, but *T. americupularis* has wider ascospores 21–25 (–26) × (–12.5) 13–15 µm; furthermore, phylogenetically it is found in different clades [7]. *T. mexicana* shares a similar distribution but differs for its smaller ascospores [(18–) 19–22 × 10–12 µm]. On the other hand, *T. poblana* has a distribution towards the central and east of Mexico (Mexico State, Puebla and Tlaxcala) and forms smaller apothecia (2–5 mm diameter), asci (170–205 × 13–15 µm), and ascospores (16–22 × 9–12 µm). *T. texcocana* differs in its ascospore size [17–21 (–22) × 11–14 (–15) µm] and shape (broadly ellipsoid).

*Tarzetta cupressicola* Sánchez-Flores, García-Jiménez, Esqueda & Raymundo, sp. nov. (Figure 4, Figure 14B and Figure 15B)

Mycobank: #851433

Diagnosis: Apothecia 5–8 mm diameter, hymenium salmon color, margin serrate to crenate, pruinose to fine-grained; ascospores 18–22 (–23) × 11–13 µm, ellipsoid, grown under *Cupressus lusitanica*.

*Type*: MEXICO. Sonora state. Yecora municipality, Los Pilares (28°23′48.9″ N, 108°47′43.6″ W), 1297 m asl, 18 September 2021, A. Gutiérrez 115 (UES10645, holotype; ENCB, isotype).

GenBank: ITS: PP825386, LSU: PP825429.

Etymology: The epithet refers to the ascocarp collected under the canopy of *Cupressus*.

Apothecia 5–8 mm in diameter, cupuliform, solitary, sessile, hymenium salmon (6A4), margin serrate to crenate, external surface pale orange (6A3), pruinose to fine-grained. Ectal excipulum 100–135 µm thick, *textura globulosa/angularis* with cells 6–19 × 5–13 µm, subhyaline, thin-walled, ectal hyphae 25–70 × 5–6 µm with 1–3 septa, hyaline. Medullary excipulum 55–85 µm thick, *textura intricata* with hyphae 3–6 µm in diameter, subhyaline. Subhymenium undifferentiated. Hymenium 250–280 µm thick. Paraphyses 2–3 µm in diameter with few septa, filiform, apex rounded, bifurcated. Asci 240–290 × 12–14 µm, cylindrical, 8-spored, uniseriate, hyaline, inamyloid. Ascospores 18–22 (–23) × 11–13 µm [x = 20 × 12 µm, *n* = 69], Q = 1.5–2, Qm = 1.75, ellipsoid, hyaline, 1–2 guttules, smooth on OM, finely rugose on SEM.

Habit: Growing between moss under the canopy of *Cupressus lusitanica* in mixed forest.

Distribution: MEXICO. It is only known from the type locality.

Material examined: Mexico, Sonora state, Yecora municipality, Los Pilares (28°23′31.5″ N, 108°47′29.8″ W), 1225 m asl, 13 September 2020, D. Madriz 52 (UES10641), 39 (UES10642); loc. cit. 17 September 2021, D. Madriz & A. Preciado 92 (UES10643), 93 (UES10644); loc. cit. (28°23′42.0″ N, 108°47′34.8″ W), 1267 m asl, 18 September 2021, D. Madriz 137 (ENCB, UES10646, paratype; ITS: PP825387, LSU: PP825430), T. Raymundo 8715 (ENCB).
Figure 4*Tarzetta cupressicola*. (**A**) Apothecia; (**B**) longitudinal section of the apothecium; (**C**) hymenium; (**D**) ectal excipulum cells; (**E**) asci and ascospores; and (**F**) ascospores.
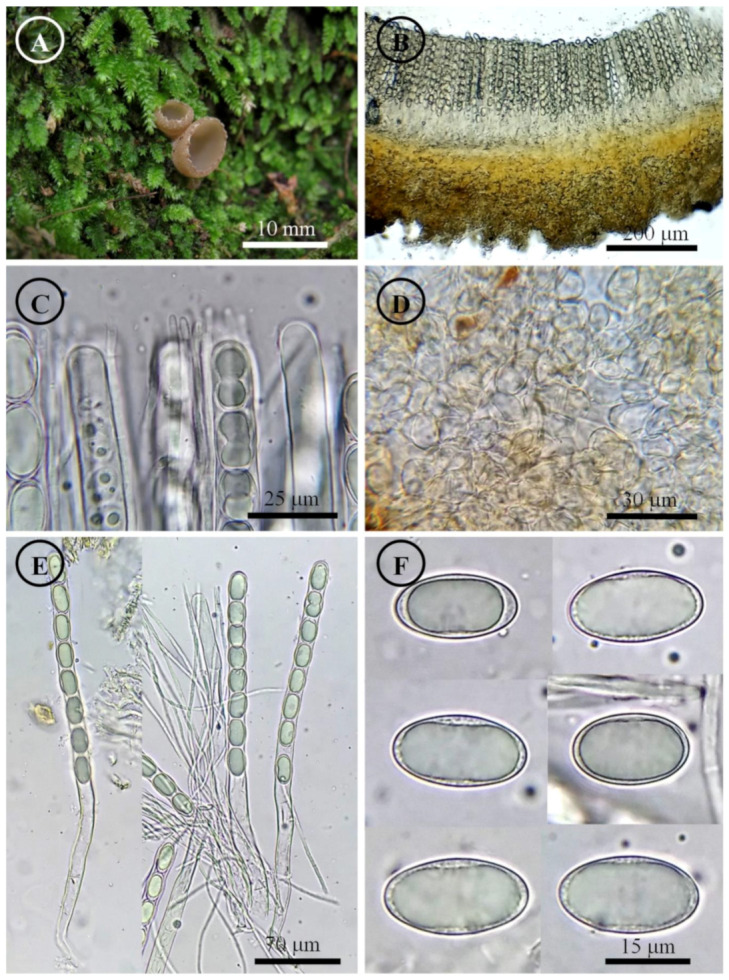



Notes: This species is characterized by forming apothecia 5–8 mm in diameter and ellipsoid ascospores [18–22 (–23) × 11–13 µm]; although it grows under the canopy of *Cupressus lusitanica*, this does not indicate that it has a mycorrhiza with this tree species. *T. texcocana* is similar in morphology but differs due to its ascospores size of 17–21 (–22) × 11–14 (–15) µm and wider paraphyses 3–4 µm in diameter. *T. durangensis* differs by having a small stipe (1.5 mm long), slightly wider paraphyses (3–4 µm), and slightly longer ascospores 20–24 × 11–13 µm.

*Tarzetta davidii* Sánchez-Flores, García-Jiménez, de la Fuente & Raymundo, sp. nov. (Figure 5, Figure 14C and Figure 15C)
Figure 5*Tarzetta davidii*. (**A**) Apothecia; (**B**) longitudinal section of the apothecium; (**C**) hymenium; (**D**) ectal excipulum cells; (**E**) asci; and (**F**) ascospores.
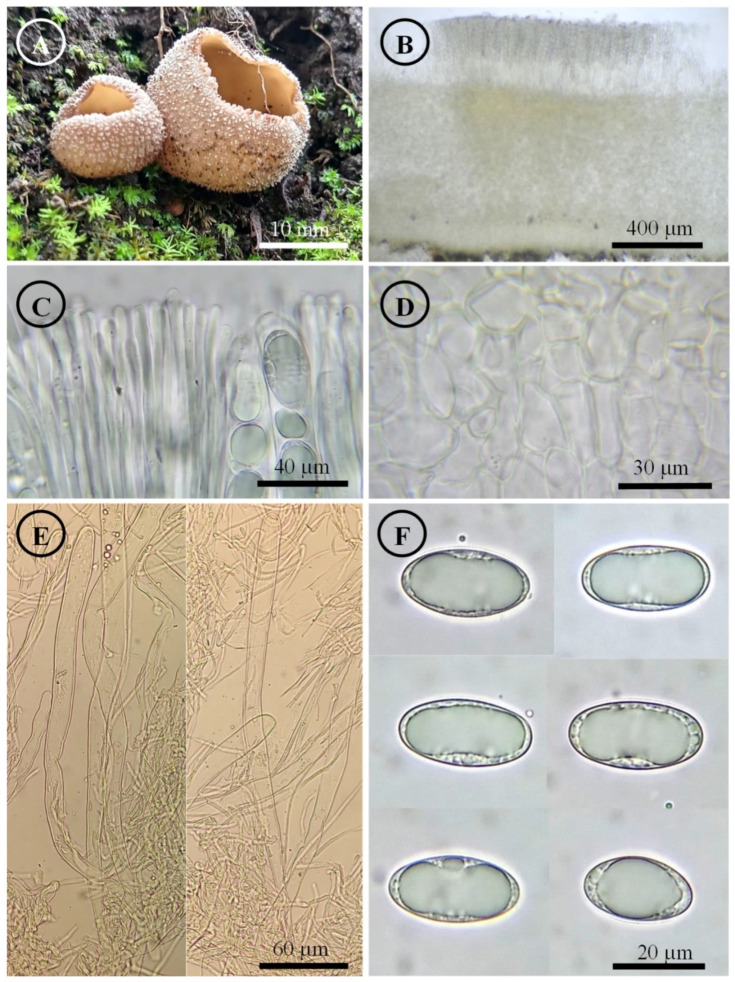



Mycobank: #851434

Diagnosis: Apothecia 15–22 mm diameter, hymenium pale orange, margin crenate to toothed, warty; ascospores (20–) 21–25 (–28) × 11–14 µm, ellipsoid to oblong, grown under *Abies religiosa*.

Type: MEXICO. Mexico State. Texcoco municipality, Mount Tlaloc (19°24′43.82″ N, 98°44′58.86″ W), 3559 m asl, 9 August 2021, D. González-Sánchez 1 (ITCV, holotype; ENCB, isotype).

GenBank: ITS: PP825390, LSU: PP825433.

Etymology: It is dedicated to David González-Sánchez, who found one of the specimens and encouraged the study of this species, nephew of M. Sánchez-Flores.

Apothecia 15–22 mm in diameter, cupuliform, solitary to scattered, sessile, hymenium pale orange (5A3), margin crenate to toothed, a cleft forms at the base of the apothecium in its connection to the substrate when it matures, external surface color greyish brown (5B3), warty. Ectal excipulum 70–230 µm thick, *textura angularis* with cells 9–37 × 6–23 µm, subhyaline, thick-walled. Medullary excipulum 543–595 µm thick, *textura intricata* with hyphae 4–7 µm in diameter, hyaline. Subhymenium undifferentiated. Hymenium 318–375 µm thick. Paraphyses 3–5 µm in diameter, filiform with septate, hyaline, apex rounded with irregular protrusions. Asci 314–350 × 15–18 µm, cylindrical, 8-spored, uniseriate, hyaline, inamyloid. Ascospores (20–) 21–25 (–28) × 11–14 µm [x = 22.8 × 12.2 µm, *n* = 57], Q = 1.5–2 (–2.2) Qm = 1.8, ellipsoid to oblong, hyaline, with a guttule covering almost the entire spore, smooth on OM, very finely rugose on SEM.

Habit: On soil, in *Abies* forest, growing under *Abies religiosa* (Kunth) Schltdl. & Cham.

Distribution: MEXICO. Transverse Neovolcanic Axis.

Material examined: Mexico, Mexico City, Álvaro Obregón town hall, Parque Nacional Desierto de los Leones, 4 September 1957, G. Guzmán 1036 (ENCB); loc. cit., 16 August 1969, B.E. Martínez Cosa 26 (ENCB); loc. cit., 29 september 1974, G. Guzmán 12019 (ENCB). Cuajimalpa town hall, Puerto Las Cruces, 10 September 1967, J. Vargas 77 (ENCB). Magdalena Contreras town hall, Los Dinamos, 20 August 1967, G. Hernández Zuñiga 95 (ENCB); loc. cit., 7 October 1974, I. García 40 (ENCB). Tlalpan town hall, Cerro Ocopiazo, 8 August 1968, G. Guzmán 6922 (ENCB). Parres, 24 July 1982, G. Rodríguez 431 (ENCB). Hidalgo state, Mineral del Monte municipality, Parque Nacional El Chico, 6 October 1974, Huerta 23 (ENCB), D. Ramos Zamora 124 (ENCB); loc. cit., 25 September 1977, G. Guzmán 16863 (ENCB), G. Vázquez 31 (ENCB); loc. cit., 22 September 1979, G. Guzmán 17859 (ENCB); loc. cit., 5 October 1980, G. Guzmán 19105 (ENCB); loc. cit., 18 July 1981, S. Acosta 618 (ENCB), 640 (ENCB); loc. cit., 1 August 1981, J. García 1597 (ITCV); loc. cit., 17 October 1982, R. Hirata 451 (ENCB), R. Valenzuela 844 (ENCB); loc. cit., 18 September 1983, R.E. Santillán 461 (ENCB). Near to Real del Monte, 23 August 1970, A. Medina López 24 (ENCB). Mineral del Chico municipality, Parque Nacional El Chico, 5 September 2023, M. Sánchez 3274 (ITCV). Mexico State, Amecameca municipality, Amecameca, 31 August 1969, J. Uribe 9 (ENCB). Donato Guerra municipality, El Capulín, 16 September 1984, L. Colón 891 (ENCB). Ixtapaluca municipality, Parque Nacional Llano Grande, 26 July 1970, M. Vázquez Hurtado 5 (ENCB). Naucalpan municipality, near to Chimalpa, 22 August 1969, J. Gimate 122-B (ENCB). Ocoyoacac municipality, Parque Nacional Insurgente Miguel Hidalgo y Costilla, July 1963, E. González 383 (ENCB); loc. cit., 13 August 1967, F. García 160 (ENCB); loc. cit., 8 September 1970, G. Guzmán 8269 (ENCB). Temascaltepec municipality, San Francisco Oxtotilpan, 21 August 1983, L. Colón 115 (ENCB). Road to Nevado de Toluca, 10 October 1977, N. Mora 120 (ENCB). Texcoco municipality, Mount Tlaloc (19°24′43.82″ N, 98°44′58.86″ W), 3559 m asl, 9 August 2021, M. Sánchez 2465 (ITCV); loc. cit., 14 November 2021, M. Sánchez 2599 (ITCV); loc. cit., 16 September 2022, M. Sánchez 2896 (ITCV), 2900 (ITCV); loc. cit., 18 September 2022, M. Sánchez 2961 (ITCV), 2971 (ITCV), L. Sánchez-Correa no number (ITCV); loc. cit., 21 September 2022, M. Sánchez 2996 (ITCV), 3016 (ITCV); loc. cit., 1 October 2023, M. Sánchez 3338 (ITCV). Zinacantepec municipality, deviation to La Peñuela, 23 July 1982, G. Guzmán 21654 (ENCB). Michoacán state, Cd. Hidalgo municipality, La Cabaña, no date, J.T. Martínez Jiménez 1 (ENCB). Pátzcuaro municipality, Los Tanques, 15 August 1980, G. Guzmán 18433 (ENCB; ITS: PP825391, LSU: PP825434). Morelos state, Huitzilac municipality, Parque Nacional Lagunas de Zempoala, 1 August 1982, R.E. Chio 374 (ENCB), G. Rodríguez 475 (ENCB). Colonia Atlixtac, 3 August 1975, G. Guzmán 12315 (ENCB). Querétaro state, Arteaga municipality, El Pingüical hill, 18 August 1990, J. García 6806 (ITCV). Tlaxcala state, Ixtenco municipality, Estación Científica La Malinche, east slope of La Malinche volcano (19°14′43.51″ N, 97°59′40.02″ W), 3166 m asl, 8 December 2018, M. Sánchez 1472 (ENCB; ITS: PP825389, LSU: PP825432), 1475 (ENCB), 1476 (ENCB, paratype; ITS: PP825388, LSU: PP825431); loc. cit., 1 October 2022, M. Sánchez 3060 (ITCV), 3067 (ITCV), 3070 (ITCV). San Pablo del Monte municipality, Parque Nacional La Malinche, 14 September 1983, González Fuentes 466 (ENCB). Tlaxco municipality, on the highway to Zacatlán, 16 September 1979, H. Matamoros 74 (ENCB).

Notes: This species is characterized by forming apothecia 15–22 mm in diameter, on which a cleft forms at the base of the apothecium in connection to the substrate when it matures, and ellipsoid to oblong ascospores [(20–) 21–25 (–28) × 11–14 µm]. It differs from *T. miquihuanensis* by the apothecia size and smaller ascospores (17–) 18–21 (–22) × 10–13 µm. *T. mesophila* is a similar species but a cleft does not form at the base of the apothecium at its connection to the substrate, while microscopically shows smaller asci 248–300 × 12–15 µm and slightly narrower ascospores (10–12 µm).

*Tarzetta durangensis* Sánchez-Flores, García-Jiménez, R. Valenz. & Raymundo, sp. nov. (Figure 6, Figure 14D and Figure 15D)

Mycobank: #851435

Diagnosis: Apothecia 4–5 mm diameter, pseudostipitate 1.5 mm, margin entire to serrated, hymenium orange, pruinose to asperulate; ascospores 20–24 × 11–13 µm, oblong.

Type: MEXICO. Durango state. Súchil municipality, Stream El Temazcal, paddock Las Alazanas, al W de la Reserva de la Biosfera de Michilía (23°27′28.57″ N, 104°14′55.10″ W), 2200 m asl, 3 September 1983, G. Rodríguez 2600 (ENCB, holotype).

GenBank: ITS: PP825392, LSU: PP825435.

Etymology: The epithet refers to the state where it was collected.

Apothecia 4–5 mm in diameter, cupuliform to discoid, solitary, with a pseudostipitate 1.5 mm long, hymenium orange (5A7), margin entire to serrated, external surface light orange (5a5) to greyish orange (5b5), pruinose to asperulate. Ectal excipulum 80–130 µm thick, *textura globulosa/angularis* with cells 13–31 × 7–18 µm, hyaline to subhyaline, thin-walled. Medullary excipulum 75–125 µm thick, *textura intricata* with hyphae 4–7 µm in diameter, hyaline. Subhymenium 55–90 µm thick. Hymenium 230–269 µm thick. Paraphyses 3–4 µm in diameter, septate, filiform, apex rounded, slightly wider, bifurcated. Asci 240–260 × 12–15 µm, cylindrical, 8-spored, uniseriate, hyaline, inamyloid. Ascospores 20–24 × 11–13 µm [x = 21.9 × 12.4 µm, *n* = 71], Q = 1.5–2.1, Qm = 1.8, oblong, hyaline, 1–2 guttules, smooth on OM, very finely rugose on SEM.

Habit: On soil, in *Pinus-Quercus* forest.

Distribution: MEXICO. It is only known from the type locality.

Material examined: Mexico, Durango state, Súchil municipality, Trampa El Olvido, Reserva de la Biosfera de Michilía, 16 August 1984, R.E. Santillán 979 (ENCB).

Notes: This species is characterized by forming apothecia 4–5 mm in diameter, with pseudostipitate, oblong ascospores (20–24 × 11–13 µm). Due to its distribution, it could be confused with *T. cupressicola*; however, this species has larger apothecia of 5–8 mm in diameter, while microscopically, it has smaller ascospores of 18–22 (–23) × 11–13 µm. *T. victoriana* is close but differs by forming smaller ascospores 17–20 × (–9) 10–11 µm.
Figure 6*Tarzetta durangensis*. (**A**) Apothecia; (**B**) longitudinal section of the apothecium; (**C**) hymenium; (**D**) ectal excipulum cells; (**E**) asci; and (**F**) ascospores.
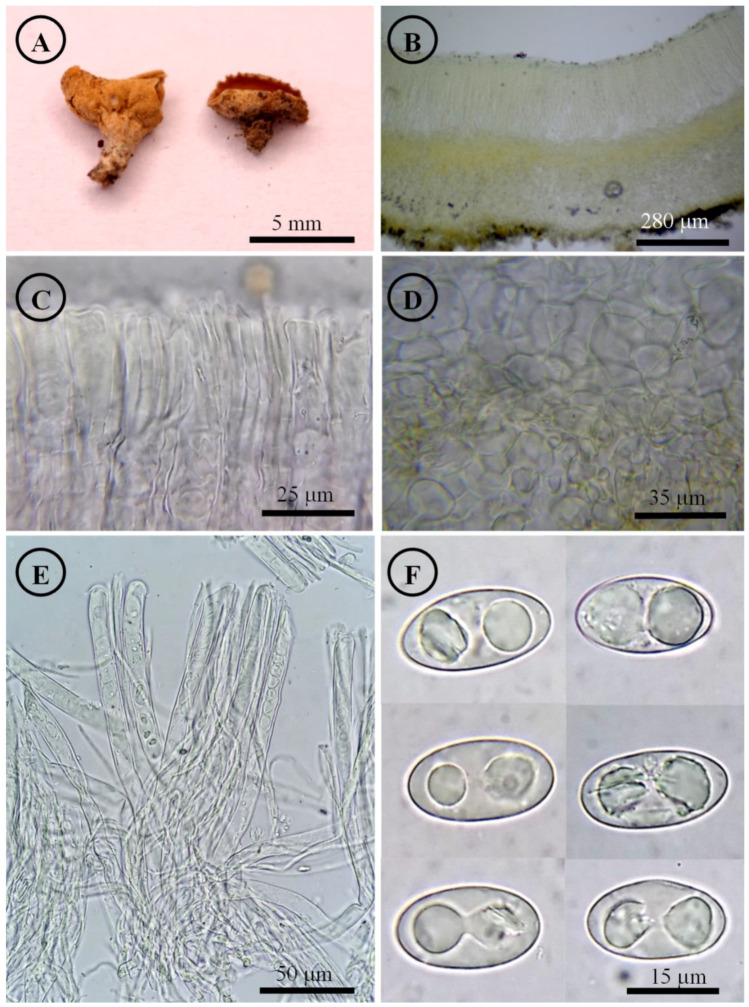



*Tarzetta mesophila* Sánchez-Flores, García-Jiménez & Raymundo, sp. nov. (Figure 7, Figure 14E and Figure 15E)

Mycobank: #851436
Figure 7*Tarzetta mesophila*. (**A**) Apothecium; (**B**) longitudinal section of the apothecium; (**C**) hymenium; (**D**) ectal excipulum cells; (**E**) asci and ascospores; and (**F**) ascospores.
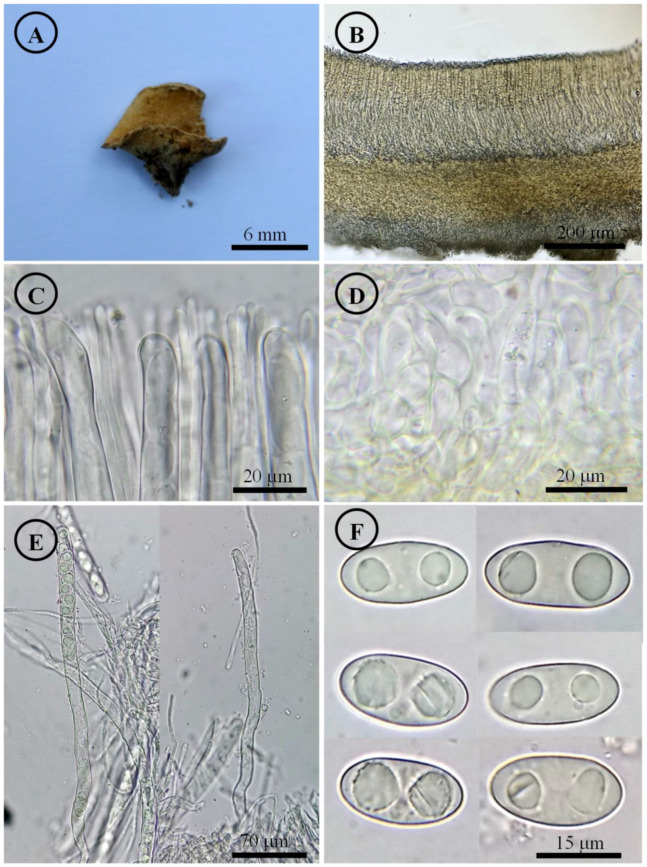



Diagnosis: Apothecia 5–9 mm diameter, hymenium light orange, margin toothed to crenate, pruinose; ascospores 19–25 (–26) × 10–12 µm, oblong to cylindrical.

Type: MEXICO. Morelos state. Huitzilac municipality, Sierra Encantada, 4 km S of Tres Marías, old México-Cuernavaca highway (19°1′10.64″ N, 99°12′59.69″ W), 10 July 1982, S. Chacón 160 (ENCB, holotype).

GenBank: ITS: PP825393, LSU: PP825436.

Etymology: The epithet refers to the “mesophyll” or tropical montane cloud forest type of vegetation.

Apothecia 5–9 mm in diameter, cupuliform, solitary to scattered, sessile, hymenium light orange (5A4), margin toothed to crenate, external surface greyish orange (5B3), pruinose to fine-grained. Ectal excipulum 115–170 µm thick, *textura angularis* with cells 16–30 × 8–18 µm, hyaline to pale yellow with thick-walled. Medullary excipulum 140–210 µm thick, *textura intricata* with hyphae 4–6 µm in diameter, hyaline. Subhymeniium undifferentiated. Hymenium 255–300 µm thick. Paraphyses 3–4 µm in diameter, filiform, septate, hyaline, apex rounded, irregular. Asci 248–300 × 12–15 µm, cylindrical, 8-spored, uniseriate, hyaline, inamyloid. Ascospores 19–25 (–26) × 10–12 µm [x = 21.7 × 10.7 µm, *n* = 71], Q = (1.7–) 1.8–2.2 (–2.4) Qm = 1.9, oblong to cylindrical, hyaline, 1–2 guttules, smooth on OM, very finely rugose on SEM.

Habit: On soil, in tropical montane cloud forest

Distribution: MEXICO. Mexico State, Morelos, and Veracruz.

Material examined: Mexico, Mexico State, Tepoztlán municipality, Sierra de Acaparrosa, 7 August 1979, G. Calderón s/n (ENCB). Morelos state, Huitzilac municipality, Tres Marías, 25 August 1967, P. Domínguez 82 (ENCB). Veracruz state, Las Vigas de Ramírez municipality, 30 October 1982, G. Guzmán 22860 (ENCB).

Notes: This species is characterized by forming apothecia 5–9 mm in diameter and oblong to cylindrical ascospores [19–25 (–26) × 10–12 µm]. It differs macroscopically from *T. davidii* by not forming a cleft at the base of the apothecium with its connection to the substrate, while microscopically, *T. mesophila* presents larger asci 248–300 × 12–15 µm and smaller ascospores unlike *T. davidii* [(20–) 21–25 (–28) × 11–14 µm]. *T. miquihuanensis* has similar morphology but is distributed in northeastern Mexico and is macroscopically larger at 33–57 mm in diameter, while microscopically, it forms smaller asci (225–335 × 9–13 µm) and ascospores [(17–) 18–21 (–22) × 10–13 µm].

*Tarzetta mexicana* Sánchez-Flores, Raymundo, de la Fuente & García-Jiménez, sp. nov. (Figure 8, Figure 14F and Figure 15F)

Mycobank: #851437

Diagnosis: Apothecia 5–17 mm diameter, hymenium whitish yellow, margin erose to crenate, granular pruinose; ascospores (18–) 19–22 × 10–12 µm, ellipsoid to oblong, grown under *Quercus* spp.

Type: MEXICO. Tamaulipas state, Miquihuana municipality, km 12 road Peña-Aserradero (23°36′03.79″ N, 99°42′22.88″ W), 2663 m asl, 03 October 2019, M. Sánchez, 1709 (ITCV, holotype; ENCB, isotype).

GenBank: ITS: PP825395, LSU: PP825438.

Etymology: The epithet refers to the country of origin of the specimens.

Apothecia 5–17 mm in diameter, cupuliform, solitary to gregarious, sessile, color orange-gray (5B2), margin erose to crenate, hymenium smooth, whitish yellow (4A2), external surface somewhat granular pruinose. Ectal excipulum 95–160 µm thick, *textura angularis* with cells 7–36 × 5–27 µm, subhyaline, ectal hyphae 3–5 µm in diameter, hyaline. Medullary excipulum 50–138 µm thick, *textura intricata* with hyphae 2.5–6 µm in diameter, hyaline. Subhymenium 25–58 µm thick. Hymenium 250–315 µm thick. Paraphyses 2–3 µm in diameter, filiform, hyaline, septate, slightly branched. Asci 220–310 × 11–14 µm, cylindrical, 8-spored, uniseriate, nailed, hyaline, inamyloid. Ascospores (18–) 19–22 × 10–12 µm [x = 19.9 × 11 µm, *n* = 74], Q = 1.5–2.1, Qm = 1.8, ellipsoid to oblong, hyaline, with a guttule that covers almost all spore, smooth on OM, finely rugose on SEM.

Habit: On soil, in forest *Quercus-Pinus*, grown under *Quercus* spp.

Distribution: MEXICO. Hidalgo, Jalisco, Puebla, Querétaro, and Tamaulipas.

Material examined: Mexico. Hidalgo state, Zacualtipan municipality, La Mojonera, 5 September 2023, M. Sánchez 3264 (ITCV). Jalisco state, road Ciudad Guzmán-San Andrés Ixtlán-El Corralito, 24 August 1974, G. Guzmán 11978 (ENCB; ITS: PP825397, LSU: PP825440). Mexico State, Iturbide municipality, Presa Iturbide, 3 October 1980, L. Guzmán-Dávalos 69 (ENCB; ITS: PP825396, LSU: PP825439). Puebla state, Atempan municipality, Canoas, soccer field, 14 September 2014, M. Sánchez 200 (FEZA; ITS: PP825398, LSU: PP825441); loc. cit., 8 October 2022, M. Sánchez, 3099 (ITCV). Capitanco hill, 7 October 2022, M. Sánchez, 3088 (ITCV), 3090 (ITCV), 3093 (ITCV). Querétaro state, Amealco municipality, Laguna de Servín (20°16′9.59″ N, 100°15′12.94″ W), 2738 m asl, 16 September 2011, J. García 18851 (ITCV, paratype; ITS: PP825394, LSU: PP825437).
Figure 8*Tarzetta mexicana*. (**A**) Apothecia; (**B**) longitudinal section of the apothecium; (**C**) hymenium; (**D**) ectal excipulum cells; (**E**) asci and ascospores; and (**F**) ascospores.
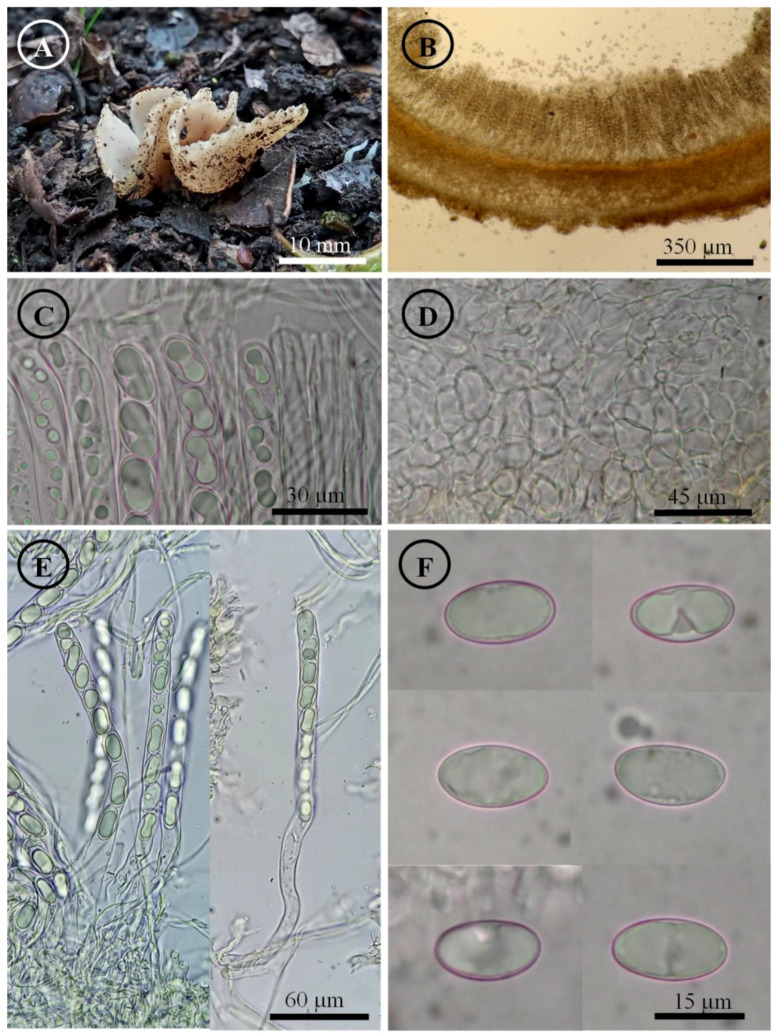



Notes: This species is characterized by forming apothecia 5–17 mm in diameter and ellipsoid ascospores [(18–) 19–22 × 10–12 µm]. *Tarzetta mexicana* differs from *T. victoriana* by having smaller ascospores (17–20 × (9–) 10–11 µm) and wider paraphyses (2–5 µm). *Tarzetta poblana* is similar in morphology but with smaller asci (170–205 × (11–) 13–15 µm) and ascospores [(16–22 × 9–12 (–13) µm]. Microscopically, it differs from *T. americupularis* by the larger ascospores of (15–) 17–25 (–26) × 9–13 (–14) µm. *Tarzetta texcocana* shows larger apothecia (4–14 mm in diameter) and wider ascospores [17–21 (–22) × 11–14 (–15) µm].

*Tarzetta miquihuanensis* Sánchez-Flores, Raymundo, Hernández-Muñoz & García-Jiménez, sp. nov. (Figure 9, Figure 14G and Figure 16A)

Mycobank: #851438

Diagnosis: Apothecia 33–57 mm diameter, hymenium pale orange, margin entire to crenate, grainy; ascospores (17–) 18–21 (–22) × 10–13 µm, ellipsoid, grown under *Quercus ariifolia* and *Q. pablillensis*.

Type: MEXICO. Tamaulipas state. Miquihuana municipality, km 12 road Peña-Aserradero, La Peña (23°36′03.79″ N, 99°42′22.88″ W), 2663 m asl, 3 October 2019, M. Sánchez 1696 (ITCV, holotype; ENCB, isotype).

GenBank: ITS: PP825399, LSU: PP825442.

Etymology: The epithet refers to the municipality where it was collected.

Apothecia 33–57 mm in diameter, cupuliform, solitary to gregarious, sessile, color pale orange (5A3), margin entire to crenate, hymenium smooth, colorless, folded to join the substrate, external surface somewhat granular pruinose whitish and grainy. Ectal excipulum 95–250 µm thick, *textura globulosa/angularis* with cells by 8–48 × 6–22 µm, subhyaline, thin wall. Medullary excipulum 240–520 µm thick, *textura intricata* with hyphae 3–5 µm in diameter, subhyaline. Subhymenium 30–75 µm thick. Hymenium 225–340 µm thick. Paraphyses 2–5 µm in diameter, filiform, hyaline, markedly septate, bifurcated. Asci 225–335 × 9–13 µm, cylindrical, 8-spored, nailed, hyaline, inamyloid. Ascospores (17–) 18–21 (–22) × 10–13 µm [x = 18.8 × 11.4 µm, *n* = 65], Q = 1.3–1.8 (–2), Qm = 1.6, ellipsoid, hyaline, one guttule that covers almost the entire spore, sometimes two guttules, smooth on OM, finely rugose on SEM.

Habit: On soil, in *Quercus-Pinus* forest, grown under *Quercus ariifolia* Trel. and *Q. pablillensis* C.H. Mull.

Distribution: MEXICO. Nuevo León and Tamaulipas.

Material examined: Mexico, Nuevo León state, Santiago municipality, El Tejocote, 18 October 1980, S. Chacón 54 (ENCB). Zaragoza municipality, El Tropezón, La Encantada, 11 July 1985, E. Cázares s/n (ITCV). Tamaulipas state, Miquihuana municipality, km 12 road Peña-Aserradero, La Peña (23°36′03.79″ N, 99°42′22.88″ W), 2663 m asl, 3 October 2014, J. García 20185 (ITCV); loc. cit., 3 October 2019, M. Sánchez 1698 (ITCV), 1705 (ITCV), 1707 (ITCV), 1710 (ITCV); loc. cit., 19 October 2019, M. Sánchez 1777 (ITCV, FEZA paratype; ITS: PP825400, LSU: PP825443). La Marcela port, 27 September 2013, J. García 19411 (ITCV).

Notes: This species is characterized by forming large apothecia 33–57 mm in diameter and ellipsoid ascospores of (17–) 18–21 (–22) × 10–13 µm. It differs from the other species of the genus by having larger apothecia and markedly septate paraphyses, even when immature. *T. davidii* is close but the apothecia is smaller (15–22 mm diameter) and the ascospores are larger (20–) 21–25 × 11–14 µm and it grows under *Abies religiosa*.
Figure 9*Tarzetta miquihuanensis*. (**A**) Apothecia; (**B**) longitudinal section of the apothecium; (**C**) hymenium; (**D**) ectal excipulum cells; (**E**) asci and ascospores; and (**F**) ascospores.
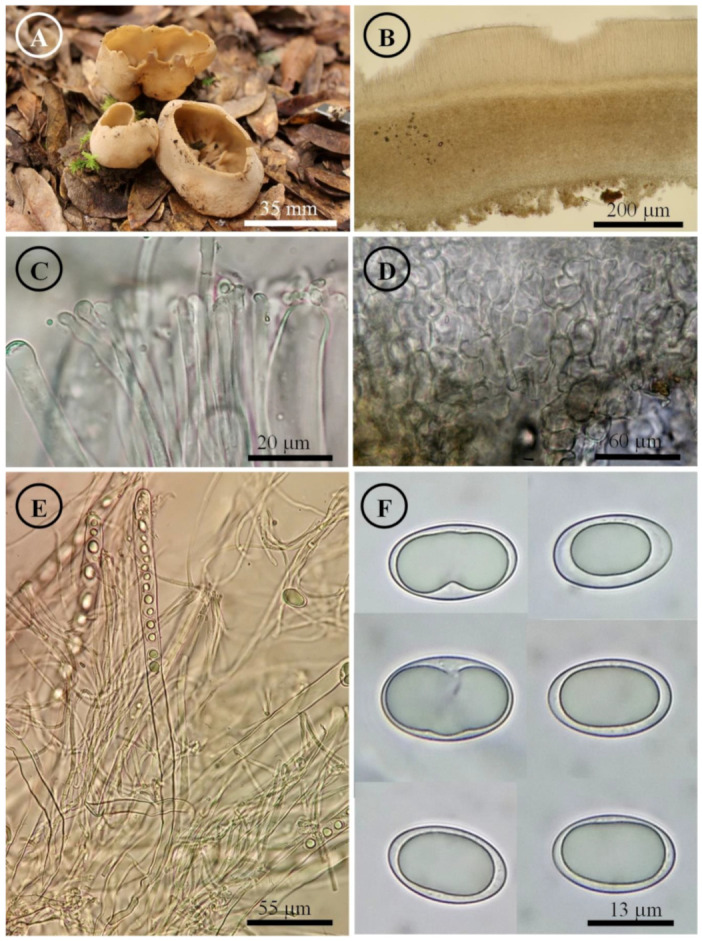



*Tarzetta poblana* Sánchez-Flores, Raymundo, Avila-Ortiz & García-Jiménez, sp. nov. (Figure 10, Figure 14H and Figure 16B)

Mycobank: #851439

Diagnosis: Apothecia 2–7 mm diameter, hymenium brownish orange, margin involute, entire to crenate, finely warty; ascospores 16–22 × 9–12 (–13) µm, ellipsoid to oblong.
Figure 10*Tarzetta poblana*. (**A**) Apothecia; (**B**) longitudinal section of the apothecium; (**C**) hymenium; (**D**) ectal excipulum cells; (**E**) asci and ascospores; and (**F**) ascospores.
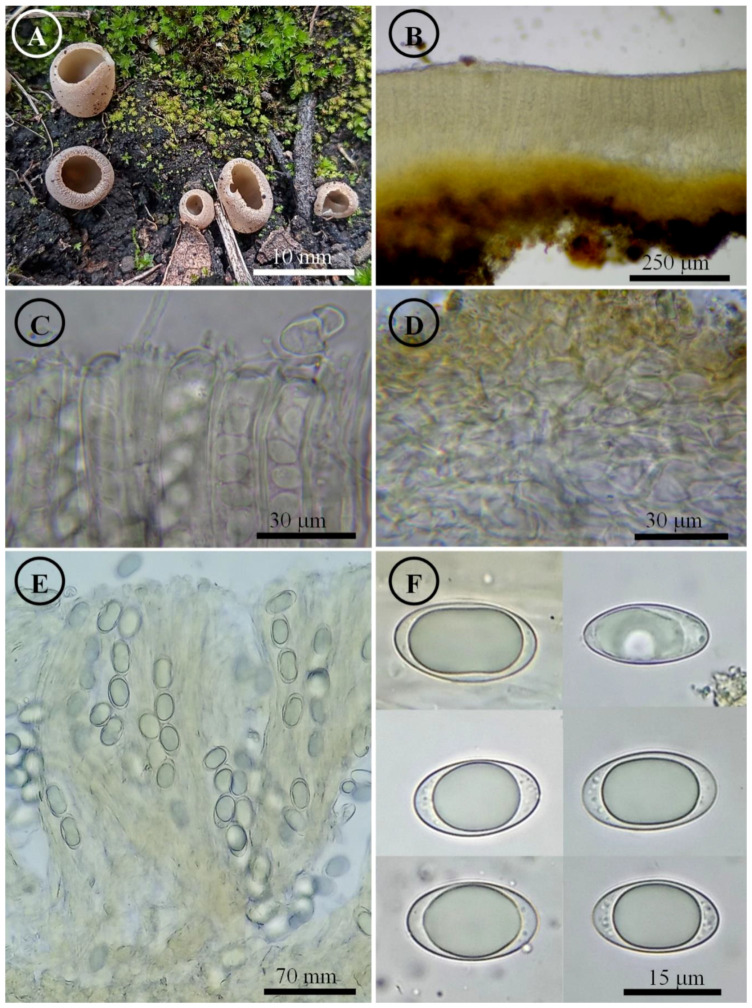



Type: MEXICO. Puebla state. Atempan municipality, Capitanco hill (19°49′1″ N, 97°27′5.43″ W), 2064 m asl, 13 September 2014, M. Sánchez 105 (FEZA, holotype; ENCB, isotype).

GenBank: ITS: PP825403, LSU: PP825446.

Etymology: The epithet refers to the state where the type specimen was collected.

Apothecia 2–7 mm in diameter, cupuliform, solitary to scattered, sessile, hymenium brownish orange (5C3) to yellowish white (4A2), margin involute, entire to crenate, lacerated when aged, external surface somewhat granular pruinose pale orange (5A3) to yellowish white (4A2), finely warty. Ectal excipulum 48–75 µm thick, *textura angularis* with cells 9–28 × 6–25 µm, hyaline, thin-walled. Medullary excipulum 60–120 µm thick, *textura intricata* with hyphae 3–6 µm in diameter, hyaline. Subhymenium 15–30 µm thick. Hymenium 180–210 µm thick. Paraphyses 2–3 µm in diameter, septate, filiform, apex rounded. Asci 170–205 × (11–) 13–15 µm, cylindrical, 8-spored, uniseriate, hyaline, inamyloid. Ascospores 16–22 × 9–12 (–13) µm [x = 19 × 11 µm, *n* = 68], Q = 1.5–2.1, Qm = 1.8, ellipsoid to oblong, hyaline, rare subcylindrical, one guttule, smooth on OM, very finely rugose on SEM.

Habit: On soil, in *Pinus-Quercus* and coniferous forest.

Distribution: MEXICO. East and central Mexico.

Material examined: Mexico, Mexico State, Amecameca municipality, Ameyalco ravine, 29 January 1980, R. Valenzuela 204 (ENCB). Road to Amecameca-Tlamacas, 23 September 1975, J. Paden 1073 (ENCB; ITS: PP825402, LSU: PP825445). Temascaltepec municipality, around of San Francisco Oxtotilpan, Parque Nacional Nevado de Toluca, 21 August 1983, L. Colón 122 (ENCB; ITS: PP825404, LSU: PP825447). Puebla state, Atempan municipality, Las Canoas, soccer field, 8 October 2022, M. Sánchez 3096 (ITCV). Juan Galindo municipality, Necaxa, August 1967, H.A. Ríos 47 (ENCB, paratype; ITS: PP825401, LSU: PP825444). Tlaxcala state, Jilotepec municipality, road Villa de Mariano, 30 September 2022, M. Sánchez 3051 (ITCV), 3052 (ITCV), 3054 (ITCV), 3055 (ITCV), 3056 (ITCV), 3058 (ITCV).

Notes: This species is characterized by forming small apothecia 2–7 mm in diameter and ellipsoid to oblong ascospores [16–22 × 9–12 (–13) µm]. *Tarzetta mexicana* is a similar species but has larger apothecia (5–17 mm), smaller ascospores of (18–) 19–22 × 10–12 µm, slightly wider paraphyses (2–4 µm), and less septate than *T. poblana*. *Tarzetta texcocana* is also similar but shows larger apothecia (4–14 mm), more crenate edge, asci of 260–280 × 12–15 µm, and slightly wider ascospores of 17–21(–22) × 11–14 (–15) µm.

*Tarzetta pseudobronca* Sánchez-Flores, García-Jiménez, Martínez-González & Raymundo, sp. nov. (Figure 11, Figure 14I and Figure 16C)

Mycobank: #851440

Diagnosis: Apothecia 13–18 mm diameter, with a pseudostipitate of 2 mm long, hymenium light orange, margin entire to crenate, pruinose to granular; ascospores 21–25 × 12–15 µm, ellipsoid to oblong, grown under *Pinus cembroides*.

Type: MEXICO. Tamaulipas state. Miquihuana municipality, Km 12 road Peña-Aserradero (23°36′03.79″ N, 99°42′22.88″ W), 2661 m asl, 3 October 2014, J. García 20240 (ITCV, holotype; ENCB, isotype).

GenBank: ITS: PP825405, LSU: PP825448.

Etymology: The epithet refers to the morphological similarity with *T. bronca* (Peck) Korf & J. K. Rogers.

Apothecia 13–18 mm in diameter, cupuliform, gregarious, with a 2 mm long pseudostipite, hymenium light orange color (5A4), margin entire to crenated that tear with maturity, external surface light brown (7D7) with pruinose to granular texture. Ectal excipulum 82.5–150 µm thick, *textura angularis* with cells 13–25 × 7–20 µm, subhyaline, thin-walled, ectal hyphae 22–33 × 5–9 µm, hyaline. Medullary excipulum 175–305 µm thick, *textura intricata* with hyphae 4–7 µm in diameter, hyaline. Subhymenium undifferentiated. Hymenium 250–255 µm thick. Paraphyses 3–6 µm in diameter, filiform, hyaline, septate, branched, slightly widening towards the apex. Asci 215–250 × 13–15 µm, cylindrical, 8-spored, uniseriate, hyaline, inamyloid. Ascospores 21–25 × 12–15 µm [x = 22.1 × 12.5 µm, *n* = 65], Q = 1.5–2, Qm = 1.7, ellipsoid to oblong, hyaline, 1–2 guttules, smooth on OM, very finely rugose on SEM.

Habit: On soil, in *Quercus-Pinus* forest, grown under *Pinus cembroides* Zucc.

Distribution: MEXICO. It is only known from the municipality of the type locality.
Figure 11*Tarzetta pseudobronca*. (**A**) Apothecia; (**B**) longitudinal section of the apothecium; (**C**) hymenium; (**D**) ectal excipulum cells; (**E**) asci and ascospores; and (**F**) ascospores.
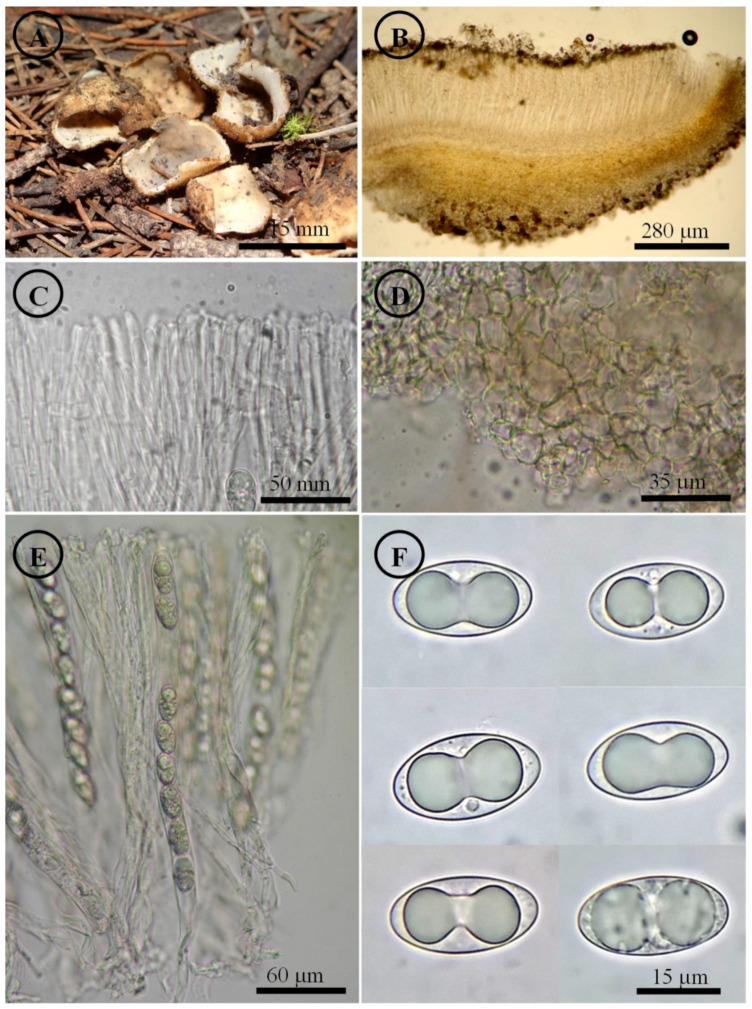



Material examined: Mexico, Tamaulipas state, Miquihuana municipality, km 6 road Peña-Aserradero, La Peña (23°34′55.31″ N, 99°42′40.03″ W), 2257 m asl, 27 September 2008, J. García 17554 (ITCV).

Notes: This species is characterized by forming small apothecia 13–18 mm in diameter and ellipsoid to oblong ascospores (21–25 × 12–15 µm). *Tarzetta bronca* is a similar species but with larger apothecia (10–35 mm in diameter), asci (250–350 × 12–16 µm), and ascospores (20–24 × 12–14 µm) [39,40] vs. *T. pseudobronca*.

*Tarzetta texcocana* Sánchez-Flores, sp. nov. (Figure 12, Figure 14J and Figure 16D)

Mycobank: #851441
Figure 12*Tarzetta texcocana*. (**A**) Apothecia; (**B**) longitudinal section of the apothecium; (**C**) hymenium; (**D**) ectal excipulum cells; (**E**) asci and ascospores; and (**F**) ascospores.
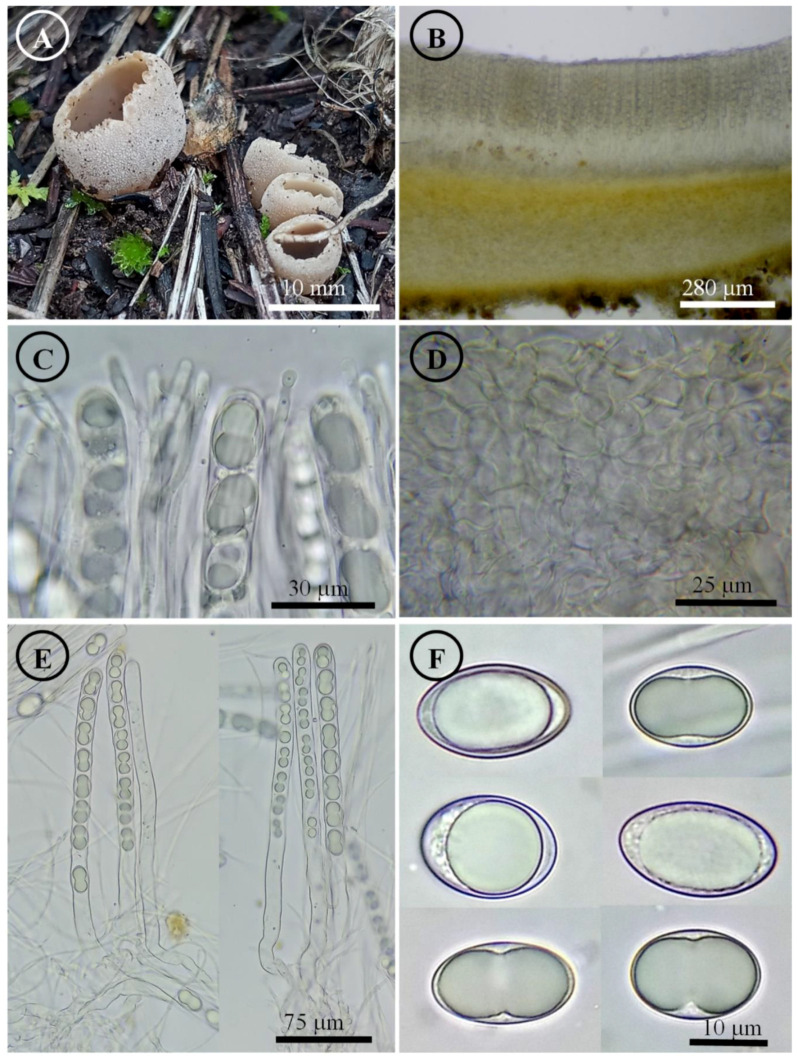



Diagnosis: Apothecia 4–14 mm diameter, hymenium greyish orange, margin entire, crenate to lacerate, grained; ascospores 17–21 (–22) × 11–14 (–15) µm, broadly ellipsoid, grown under *Quercus* spp.

Type: MEXICO. Mexico state, Texcoco municipality, Mount Tlaloc (19°24′43.82″ N, 98°44′58.86″ W), 3559 m asl, 14 November 2021, M. Sánchez 2585 (ITCV, holotype; ENCB, FEZA, isotype).

GenBank: ITS: PP825407, LSU: PP825450.

Etymology: The epithet refers to the municipality of origin of the specimens.

Apothecia 4–14 mm in diameter, cupuliform when young to discoid when old, solitary to scattered, sessile, hymenium greyish orange (5B3), margin entire, crenate to lacerate, external surface orange-white (5A2) to pale orange (5A3), grained. Ectal excipulum 65–115 µm thick, *textura angularis* with cells 11–30 × 8–20 µm, hyaline, thin wall, terminals hyphae 14–31 × 5–6 µm, hyaline. Medullary excipulum 145–210 µm thick, *textura intricata* with hyphae 3–7 µm in diameter, hyaline. Subhymenium 63–100 µm thick. Hymenium 255–290 µm thick. Paraphyses 3–4 µm in diameter, filiform, septate, apex rounded, nodulose, bifurcated at the base. Asci 260–280 × 12–15 µm, cylindrical, 8-spored, uniseriate, hyaline, inamyloid. Ascospores 17–21 (–22) × 11–14 (–15) µm [x = 19.5 × 13 µm, *n* = 64], Q = 1.2–1.7, Qm = 1.4, broadly ellipsoid, hyaline, one guttule that covers almost the entire ascospore, smooth on OM, finely rugose on SEM.

Habit: On soil, in mixed forest, grown under *Quercus* spp.

Distribution: MEXICO. It is only known from the type locality.

Material examined: Mexico, Mexico State, Texcoco municipality, Mount Tlaloc (19°24′43.82″ N, 98°44′58.86″ W), 3559 m asl, 14 November 2021, M. Sánchez 2576 (ITCV, FEZA, paratype; ITS: PP825406, LSU: PP825449), 2587 (ITCV), 2590 (ITCV), 2593 (ITCV), 2597 (ITCV); loc. cit., 16 September 2022, M. Sánchez 2890 (ITCV), 2894 (ITCV), 2902 (ITCV), 2904 (ITCV), 2906 (ITCV), 2910 (ITCV), 2912 (ITCV); loc. cit., 18 September 2022, M. Sánchez 2977 (ITCV); loc. cit., 21 September 2022, M. Sánchez 2994 (ITCV), 2999 (ITCV), 3002 (ITCV), 3007 (ITCV), 3012 (ITCV); loc. cit., 4 October 2022, M. Sánchez 3079 (ITCV); loc. cit., 1 October 2023, M. Sánchez 3339 (ITCV); loc. cit., 4 October 2023, M. Sánchez 3374 (ITCV).

***Notes***: This species is characterized by forming small apothecia 4–14 mm in diameter, and broadly ellipsoid ascospores [17–21 (–22) × 11–14 (–15) µm]. *Tarzetta cupressicola* is a similar species but with narrower paraphyses (2–3 µm) and ascospores of 18–22 × 11–13 µm and is associated with *Cupressus lusitanica*. *Tarzetta poblana* is a similar species but with apothecia smaller than 2–7 mm, narrower paraphyses (2–3 µm), asci of 170–205 × 13–15 µm, and ascospores of 17–21 (–22) × 9–12 µm. *Tarzetta mexicana* differs by forming narrower ascospores of (18–) 19–22 × 10–12 µm.

*Tarzetta victoriana* Sánchez-Flores, Raymundo, Hernández-Del Valle & García-Jiménez, sp. nov. (Figure 13, Figure 14K,L and Figure 16E,F)

Mycobank: #851442

Diagnosis: Apothecia 2–25 mm diameter, hymenium pale orange, margin involute to crenate, pruinose to grainy; ascospores 17–20 × (9–) 10–11 µm, ellipsoid, grown under *Quercus rysophylla* and *Q. polymorpha*.

Type: MEXICO. Tamaulipas state. Victoria municipality, Puerto El Paraíso community (23°31′38.99″ N, 99°12′20.04″ W), 1650 m asl, 17 October 2019, M. Sánchez 1752 (ITCV, holotype; ENCB, isotype).

GenBank: ITS: PP825409, LSU: PP825452.

Etymology: The epithet refers to the municipality where it was collected and described for the first time.

Apothecia 2–25 mm in diameter, cupuliform, solitary to gregarious, sessile, sometimes pseudostipitate, hymenium pale orange (5A3) to brownish orange (5C3), margin involute to crenate, hymenium smooth, external surface greyish orange color (5B4) to brownish orange (5C3), pruinose to grainy. Ectal excipulum 55–125 µm thick, *textura globulosa*/*angularis* with cells 10–33 × 7–26 µm, subhyaline, with ectal hyphae of 3–6 µm in diameter, hyaline. Medullary excipulum 85–128 µm thick, *textura intricata* with hyphae 2–5 µm in diameter, subhyaline. Subhymenium undifferentiated. Hymenium 225–295 µm thick. Paraphyses 2–5 µm in diameter, filiform, hyaline, septate, branched, with the apex slightly widened and irregular. Asci 225–280 × 11–13 µm, cylindrical, 8-spored, nailed, hyaline, inamyloid. Ascospores 17–20 × (9–) 10–11 µm [x = 18.8 × 10.2 µm, *n* = 66], Q = 1.6–2 (–2.1), Qm = 1.8, ellipsoid, hyaline, one guttule spanning all spore, smooth on OM, finely rugose on SEM.
Figure 13*Tarzzetta victoriana*. (**A**) Apothecia; (**B**) longitudinal section of the apothecium; (**C**) hymenium; (**D**) ectal excipulum cells; (**E**) asci and ascospores; and (**F**) ascospores.
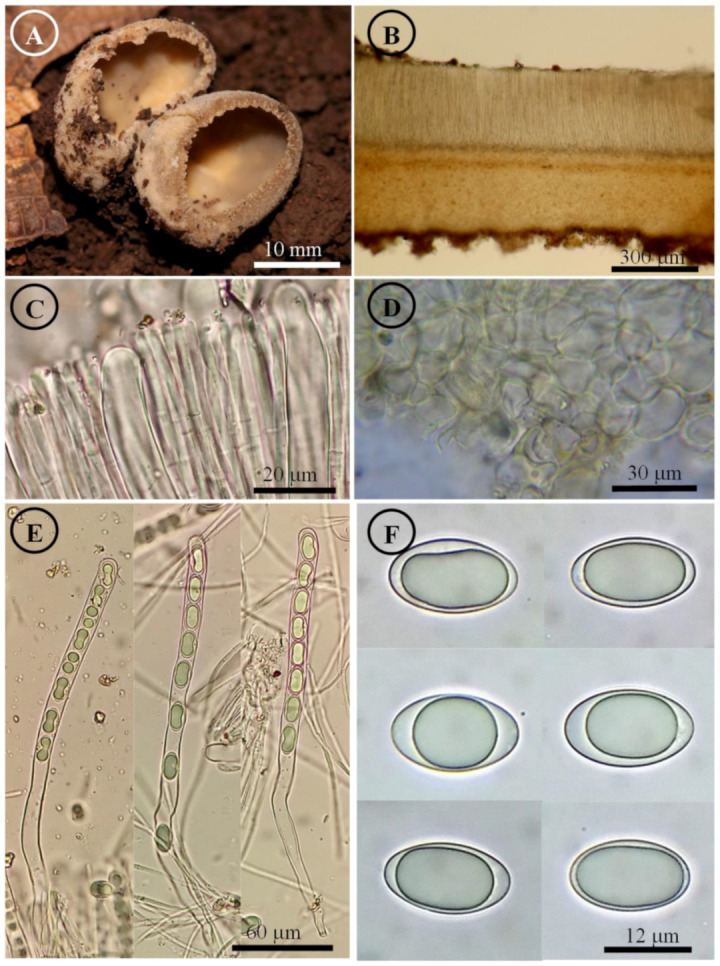

Figure 14Ascospores on OM. (**A**) *Tarzetta americupularis*; (**B**) *T. cupressicola*; (**C**) *T. davidii*; (**D**) *T. durangensis*; (**E**) *T. mesophile*; (**F**) *T. mexicana*; (**G**) *T. miquihuanensis*; (**H**) *T. poblana*; (**I**) *T. pseudobronca*; (**J**) *T. texcocana*; and (**K,L**) *T. victoriana*.
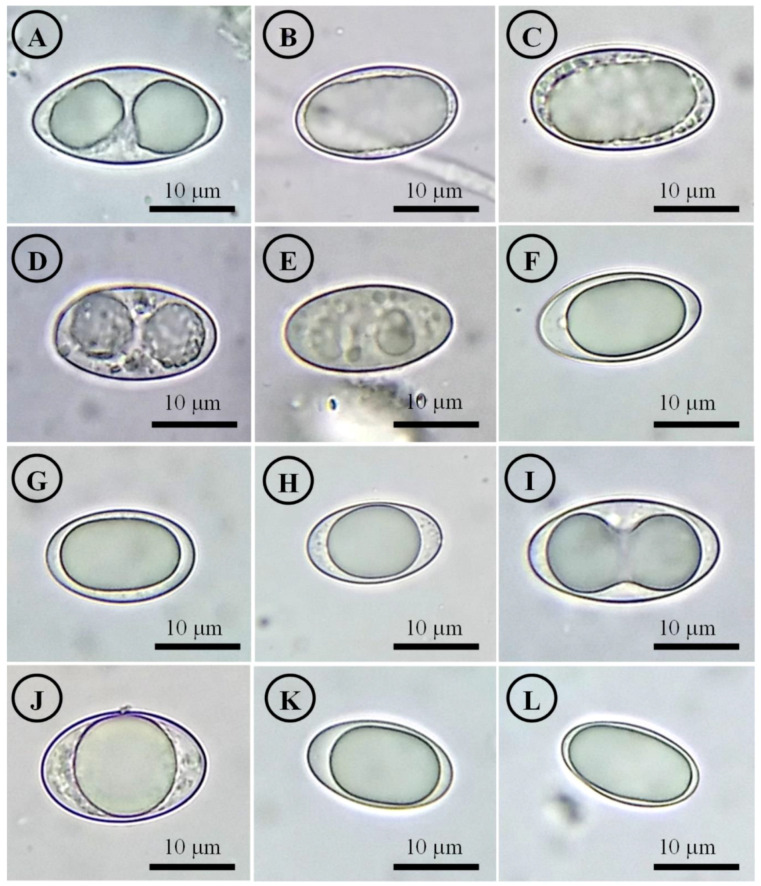



Habit: On soil, in *Quercus* spp. forest grown under *Quercus rysophylla* Weath and *Q. polymorpha* Schltdl. & Cham.

Distribution: MEXICO. It is only known from the state of Tamaulipas.

Material examined: Mexico, Tamaulipas state, Jaumave municipality, Sierra Madre, 8 November 2019, M. Sánchez 1862 (ITCV), 1863 (ITCV). Victoria municipality, Puerto El Paraíso community (23°31′38.99″ N, 99°12′20.04″ W), 1650 m asl, 17 October 2019, M. Sánchez 1754 (ITCV), 1768 (ITCV), 1773 (ITCV); loc. cit., 1 November 2019, M. Sánchez 1830 (ITCV), 1845 (ITCV); loc. cit., 21 November 2019, M. Sánchez 1923 (ITCV), 1928 (ITCV); loc. cit., 4 September 2020, M. Sánchez 2080 (ITCV); loc. cit., 13 September 2020, M. Sánchez 2100 (ITCV), 2115 (ITCV, ENCB, paratype; ITS: PP825408, LSU: PP825451), 2120 (ITCV); El Madroño, km 19 road Ciudad Victoria-Tula (23°36′16.32″ N, 99°13′45.20″ W), 1443 m asl, 11 November 2019, M. Sánchez 1883 (ITCV), 1896 (ITCV), 1898 (ITCV); Las Mulas, 26 August 2020, M. Sánchez 2075 (ITCV).

Notes: This species is characterized by forming small apothecia 2–25 mm in diameter and ellipsoid ascospores [17–20 × (9–) 10–11 µm]. The microscopic differences among *T. pseudobronca*, *T. victoriana*, and *T. mexicana* are the spore length and width; *T. victoriana* has the smallest and narrowest spores, while *T. pseudobronca* has the largest and widest spores (21–25 × 12–15 µm). *T. mexicana* has smaller and narrower ascospores (18–) 19–22 × 10–12 µm than *T. pseudobronca*.
Figure 15Ascospores on SEM. (**A**) *Tarzetta americupularis*; (**B**) *T. cupressicola*; (**C**). *T. davidii*; (**D**) *T. durangensis*; (**E**) *T. mesophila*; and (**F**). *T. mexicana*.
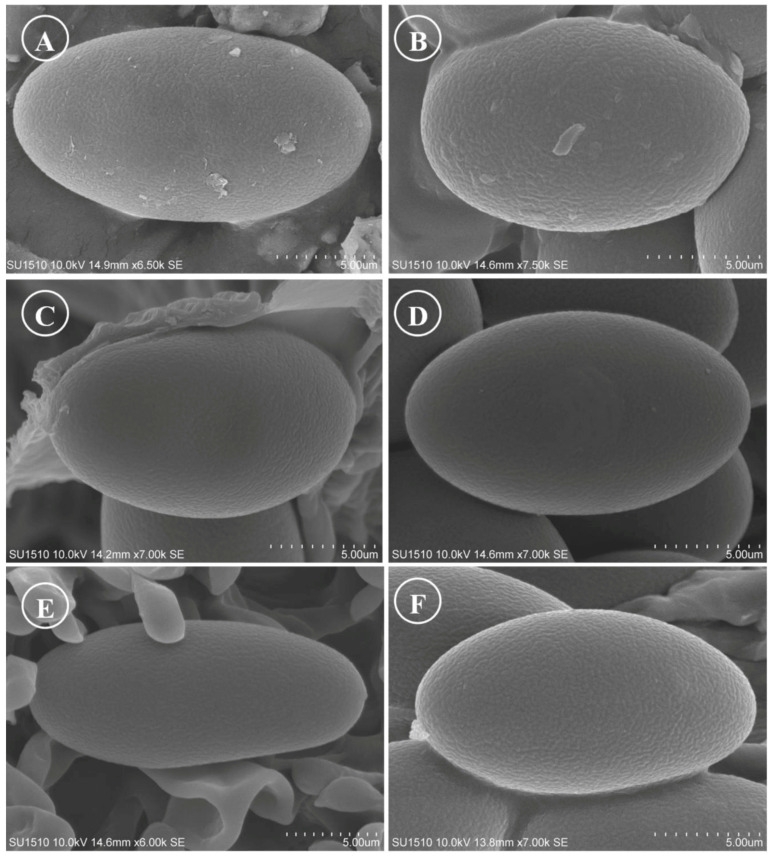

Figure 16Ascospores on SEM. (**A**) *Tarzetta miquihuanensis*; (**B**) *T. poblana*; (**C**) *T. pseudobronca*; (**D**) *T. texcocana*; and (**E**,**F**) *T. victoriana*.
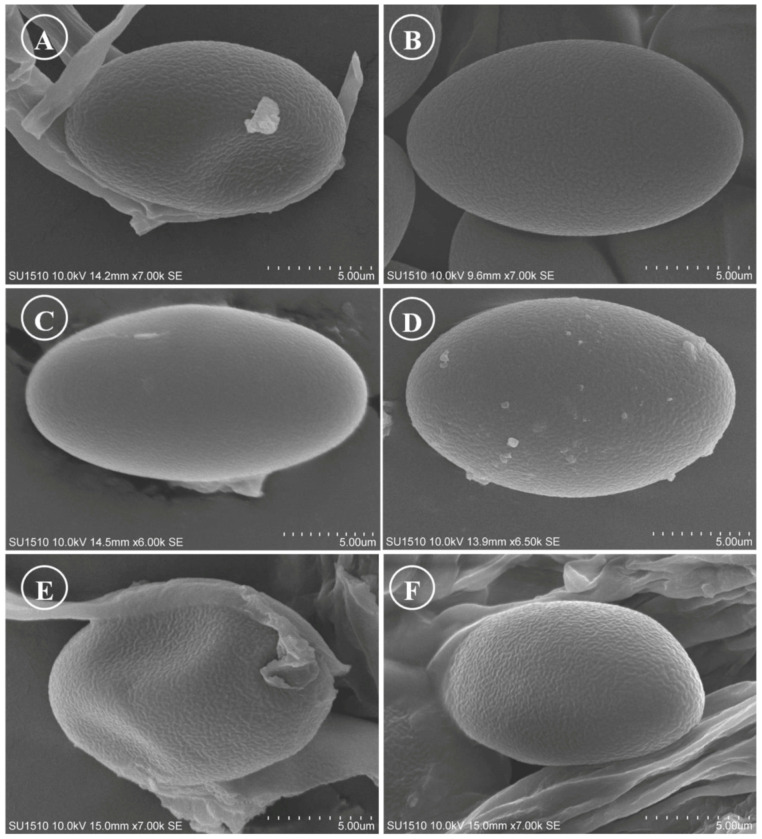



**Table 2 jof-10-00403-t002:** Comparison of *Tarzetta* species.

*Tarzetta*	Country	Vegetation	Apothecia, Diameter, and Color	Margin and Edge of the Apothecia	Paraphysis	Asci	Ascospores
***T. americupularis*** Sánchez-Flores, García-Jiménez, R. Valenz. & Raymundo	México (This study)	*Pinus-Quercus* and coniferous forest	4–8 mm diameter, cupuliform, greyish orange to pale orange color	Crenate	2–4 µm, filiform, rounded to abrupt apex	190–310 (–325) × 12–15 µm	(15–) 17–25 (–26) × 9–13 (–14) µm, ellipsoid to oblong, very finely rugose on SEM
***T. bronca*** (Peck) Korf & J. K. Rogers	USA [39,40]	Not indicated	10–35 mm diameter, cupuliform, deep olive beige to beige-yellow color.	Crenulated	Size not indicated, septate, simple, forked to branched, gnarled to claviform towards the apex	250–350 × 12–16 µm	20–24 × 12–14 µm, ellipsoid, smooth
***T. catinus*** (Holmsk.) Korf & J. K. Rogers	England [41]	Not indicated	40–50 mm diameter	Not indicated	Not indicated	Not indicated	(19.5–) 20–24 × 11–14 µm, ellipsoid, smooth
***T. cupressicola*** Sánchez-Flores, García-Jiménez, Esqueda & Raymundo	Mexico (This study)	Mixed forest	5–8 mm diameter, greyish brown to yellowish brown color	Sawn to slightly crenate	2–3 µm, filiform, apex rounded, little septate, forked	240–290 × 12–14 µm	18–22 (–23) × 11–13 µm, ellipsoid, finely rugose on SEM
***T. cupularis*** (L.) Lambotte	England [41]	Not indicated	<20 mm diameter	Not indicated	Not indicated	Not indicated	21–25 (–26) × (12.5–) 13–15 µm, ellipsoid to broadly ellipsoid, smooth
***T. davidii*** Sánchez-Flores, García-Jiménez & Raymundo	Mexico (This study)	*Abies* forest	15–22 mm diameter, cupuliform, greyish brown to pale orange color	Crenate to dentate	3–5 µm, filiform, deeply septate, apex rounded, with irregular protuberances.	314–350 × 15–18 µm	(20–) 21–25 (–28) × 11–14 µm, ellipsoid to oblong, very finely rugose on SEM
***T. durangensis*** Sánchez-Flores, García-Jiménez, R. Valenz. & Raymundo	Mexico (This study)	*Pinus-Quercus* forest	4–5 mm diameter, cupuliform to discoid, light orange, greyish orange to orange color, with a pseudostipitate	Sawed	3–4 µm, filiform, septate, slightly forked, apex rounded	240–260 × 12–15 µm	20–24 × 11–13 µm, oblong very finely rugose on SEM
***T. mesophila*** Sánchez-Flores, García-Jiménez & Raymundo	Mexico (This study)	Tropical montane cloud forest	5–9 mm diameter, cupuliform, greyish orange to light orange color	toothed to crenate	3–4 µm, filiform, septate, apex rounded to rarely irregular	248–300 × 12–15 µm	19–25 (–26) × 10–12 µm, oblong to subcylindrical, very finely rugose on SEM
***T. mexicana*** Sánchez-Flores, Raymundo, de la Fuente & García-Jiménez	México (This study)	*Quercus-Pinus* forest	5–17 mm diameter, cupuliform, greyish orange to yellowish white color	erose to crenate	2–3 µm, filiform, septate, slightly branched	220–310 × 11–14 µm.	(18–) 19–22 × 10–12 µm, ellipsoid to oblong, finely rugose on SEM
***T. miquihuanensis*** Sánchez-Flores, Raymundo, Hernández-Muñoz & García-Jiménez	México (This study)	*Quercus-Pinus* forest	33–57 mm diameter, cupuliform, pale orange color	Entire to crenate	2–5 µm, filiform, deeply septate, bifurcated	225–335 × 9–13 µm.	(17–) 18–21 (–22) × 10–13 µm, ellipsoid, finely rugose on SEM
***T. poblana*** Sánchez-Flores, Raymundo, Avila-Ortiz & García-Jiménez	Mexico (This study)	*Pinus-Quercus* and coniferous forest	2–7 mm diameter, pale orange, cupuliform, brownish orange to yellowish white color	Involute, entire to crenate	2–3 µm, filiform, septate, apex rounded	170–205 × (11–) 13–15 µm	16–22 × 9–12 (–13) µm, ellipsoid to oblong, very finely rugose on SEM
***T. pseudobronca*** Sánchez-Flores, García-Jiménez, Martínez-González & Raymundo	México (This study)	*Quercus-Pinus* and *Pinus cembroides* forest	13–18 mm diameter, cupuliform, greyish brown to light orange color, with a pseudostipitate	Entire to crenate	3–6 µm, filiform, septate, branched, widening towards the apex	215–250 × 13–15 µm	21–25 × 12–15 µm, ellipsoid to oblong, very finely rugose on SEM
***T. texcocana*** Sánchez-Flores	Mexico (This study)	Coniferous forest	4–14 mm diameter, cupuliform, orange white, pale orange to greyish brown color	Entire to crenate	3–4 µm, filiform, scarcely septate	260–280 × 12–15 µm	17–21 (–22) × 11–14 (–15) µm, broadly ellipsoid, finely rugose on SEM
***T. victoriana*** Sánchez-Flores, Raymundo, Hernández-del Valle & García-Jiménez	México (This study)	*Quercus* spp. forest	2–25 mm diameter, cupuliform, greyish orange, brownish orange, pale orange to brownish orange color, sessile to pseudostipitate	Crenate	2–5 µm, filiform, septate, branched, with the apex slightly widened to irregular	225–280 × 11–13 µm	17–20 × (9–) 10–11 µm, ellipsoid, finely rugose on SEM


**Key to *Tarzetta* species from Mexico**
1. Apothecia growing in coniferous or mixed forests in temperate regions21. Apothecia growing in *Quercus* forests or tropical montane cloud forests82. Apothecia growing in coniferous forests 32. Apothecia growing in mixed forests 53. Apothecia less than 10 mm with margin crenate, color salmon grown under *Cupressus lusitanica*
*T. cupressicola*
3. Apothecia larger than 10 mm 44. Apothecia associated with *Pinus cembroides*, excipule light brown
*T. pseudobronca*
4. Apothecia associated with *Abies religiosa*, excipule greyish brown warty
*T. davidii*
5. Apothecia with external surface warty 65. Apothecia with external surface asperulate76. Apothecia color pale orange margin entire, involute to crenate, ascospores 16–22 × 9–12 (–13) µm, ellipsoid-oblong 
*T. poblana*
6. Apothecia color greyish orange with margin crenate, ascospores 17–21 (–22) × 11–14 (–15) µm, broadly ellipsoid 
*T. texcocana*
7. Apothecia substipitate, ascospores 20–24 × 11–13 µm, oblong with blunt ends
*T. durangensis*
7. Apothecia sessile, ascospores (15–) 17–25 (–26) × 9–13 (–14) µm, ellipsoid to oblong with acute ends 
*T. americupularis*
8. Apothecia growing in tropical montane cloud forest, ascospores 19–25 (–26) × 10–12 µm, subcylindrical
*T. mesophila*
8. Apothecia growing in *Quercus* forests, ascospores ellipsoid99. Apothecia larger than 30 mm, margin entire to crenate
*T. miquihuanensis*
9. Apothecia small, less than 25 mm, margin crenate to eroded1010. Apothecia substipitate, color pale orange, margin involute crenate, associated with *Quercus rysophylla* and *Q. polymorpha*
*T. victoriana*
10. Apothecia sessile, color orange-gray to whitish, margin eroded, associated with several species of *Quercus*
*T. mexicana*



## 4. Discussion

In this study, 11 species of *Tarzetta* are described from Mexico, based on ecological, morphological, and molecular data. Combined analyses of two datasets (ITS and LSU) showed two strongly supported clades described as follows: Clade I, with most of the species growing on the Sierra Madre Oriental, is mainly associated with *Quercus* species, except *T. davidii*, which is putatively associated with *Abies religiosa* at least within the Transversal Neovolcanic Axis [42]. Species of this clade show paraphyses generally slightly branched, filiform, and septate. Clade II, with most of the species growing on the Sierra Madre Occidental, is associated with conifers; *T. texcocana* and *T. pseudobronca* seem to be the exception, where the former may be associated with *Quercus* species, while the second associated with *Pinus cembroides*. Most of the species of this clade have paraphyses bifurcate with few septa; nevertheless, *T. pseudobronca* can show branched paraphysis.

According to Van Vooren et al. [7], the main diagnostic characteristics for the description of *Tarzetta* species are the size of the ascoma and ascospores, as well as the host. In this work, two clades are observed, separated mainly by the paraphyses structure. Van Vooren et al. [7] considered this character of little taxonomic value because the apical area of the paraphyses seems to be highly variable in the same species, depending on the development stage of the area where the observations are made. However, there is a tendency for Mexican species to group, according to this character, although it is not consistent. According to the host, a marked separation of the clades is observed, with the species of Clades I and II mostly associated with conifers and *Quercus* species, respectively. In the case of *T. cupressicola*, this species grows under *Cupressus lusitanica*; however, there is no evidence that it is directly associated with mycorrhiza.

Although most species of this genus have been described and cited with smooth spores, except *J. jafneospora* with verrucose ascospores [7,9,39], all Mexican species seen in OM are smooth but under SEM they show finely rugose ornamentation (Figure 14 and Figure 15). Likewise, it is suggested to carefully review European and American species under SEM to confirm the smooth ascospores.

## 5. Conclusions

Our results obtained in this study confirm that *T. catinus* and *T. cupularis* are not distributed in the American continent and they are restricted to Europe, as mentioned by Van Vooren et al. [6]. Therefore, it is inferred that the genus Tarzetta is an open field with more taxa for the American continent, waiting to be described. In the case of *T. brasiliensis* and *T. microspora*, these taxa must be reviewed due to the lack of information, and their taxonomic position must also be validated. *Tarzetta* species are ectomycorrhizal, where the species described in our study are mainly associated with *Abies*, *Pinus*, and *Quercus* in the temperate forests of Mexico, showing finely rugose ascospores and ornamentation not previously reported in the genus.

## Figures and Tables

**Table 1 jof-10-00403-t001:** Taxa information and GenBank accession numbers of the sequences used in this study.

Species Name	Voucher Number	GenBank Accession	
		ITS	LSU
*Hypotarzetta insignis*	LY:NV 2014-03-07	MN712290	MN712245
*Tarzetta alnicola*	LY:NV 2017-08-33	MN712300	MN712255
*Tarzetta alnicola*	LY:NV 2017-08-36	MN712301	MN712256
*Tarzetta alpina*	LY:NV 2009-08-11	-----	MN712259
*Tarzetta americupularis*	T. Raymundo 3833 Type ENCB	PP825384	PP825427
*Tarzetta americupularis*	E. García 310 ENCB	PP825385	PP825428
*Tarzetta betulicola*	M.P. 2018-133	-----	MN712273
*Tarzetta bronca*	SAK-17-0423-5	MN712304	MN712261
*Tarzetta catinus*	LY:NV 2002-06-05	-----	MN712277
*Tarzetta catinus*	V.R. 20190509	MN712315	MN712274
*Tarzetta* cf. *catinus*	M.C. 18-10-13	MN712303	MN712258
*Tarzetta confusa*	HKAS 115755	NR175743	NG081522
*Tarzetta confusa*	HKAS 115756	MZ438005	MZ438008
*Tarzetta cupularis*	LY:NV 2019-05-11	MN712319	MN712280
*Tarzetta cupularis*	LY:NV 2006-10-31	MN712322	MN712283
*Tarzetta cupressicola*	A. Gutiérrez 115 Type UES	PP825386	PP825429
*Tarzetta cupressicola*	D. Madriz 137 UES	PP825387	PP825430
*Tarzetta davidii*	M. Sánchez 1476 ENCB	PP825388	PP825431
*Tarzetta davidii*	M. Sánchez 1472 ENCB	PP825389	PP825432
*Tarzetta davidii*	D. González-Sánchez 01 Type ITCV	PP825390	PP825433
*Tarzetta davidii*	G. Guzmán 18433 ENCB	PP825391	PP825434
*Tarzetta durangensis*	G. Rodríguez 2600 Type ENCB	PP825392	PP825435
*Tarzetta gaillardiana*	ALL9409	-----	DQ220439
*Tarzetta gregaria*	LY:NV 2017-08-16	MN712288	MN712243
*Tarzetta jafneospora*	JAC12299	-----	MK431445
*Tarzetta linzhiensis*	HKAS 115754	MZ438004	MZ438007
*Tarzetta mesophila*	S. Chacón 160 Type ENCB	PP825393	PP825436
*Tarzetta melitensis*	LY:CS871	MN712324	MN712285
*Tarzetta mexicana*	J. García 18851 ITCV	PP825394	PP825437
*Tarzetta mexicana*	M. Sánchez 1709 Type ITCV	PP825395	PP825438
*Tarzetta mexicana*	L. Guzmán-Dávalos 69 ENCB	PP825396	PP825439
*Tarzetta mexicana*	G. Guzmán 11978 ENCB	PP825397	PP825440
*Tarzetta mexicana*	M. Sánchez 200 FEZA	PP825398	PP825441
*Tarzetta miquihuanensis*	M. Sánchez 1696 Type ITCV	PP825399	PP825442
*Tarzetta miquihuanensis*	M. Sánchez 1777 ITCV	PP825400	PP825443
*Tarzetta oblongispora*	TUR-A 216571	OR243636	-----
*Tarzetta oblongispora*	TUR-A 216572	OR243629	-----
*Tarzetta oblongispora*	TUR-A 216574	OR243631	MN712271
*Tarzetta ochracea*	LY:NV 2018-06-11	MN712313	MN712269
*Tarzetta ochracea*	LY:NV 2019-06-01 Type	MN712310	MN712268
*Tarzetta poblana*	H.A. Ríos 47 ENCB	PP825401	PP825444
*Tarzetta poblana*	J. Paden 1073 ENCB	PP825402	PP825445
*Tarzetta poblana*	M. Sánchez 105 Type FEZA	PP825403	PP825446
*Tarzetta poblana*	L. Colón 122 ENCB	PP825404	PP825447
*Tarzetta pusilla*	KH0366	-----	DQ220440
*Tarzetta pseudobronca*	J. García 20240 Type ITCV	PP825405	PP825448
*Tarzetta pseudocatinus*	LY:NV 2014-08-19	MN712295	MN712250
*Tarzetta pseudocatinus*	LY:NV 2019-05-12	MN712294	MN712249
*Tarzetta spurcata*	AMNH44124	-----	DQ220441
*Tarzetta texcocana*	M. Sánchez 2576 ITCV	PP825406	PP825449
*Tarzetta texcocana*	M. Sánchez 2585 Type ITCV	PP825407	PP825450
*Tarzetta tibetensis*	HKAS 127118	OQ422965	OQ418059
*Tarzetta tibetensis*	HKAS 127117	OQ417936	OQ418058
*Tarzetta quercus-ilicis*	LY:NV 2014-03-20	MN712306	MN712264
*Tarzetta quercus-ilicis*	M.C. 14-3-15	MN712307	MN712265
*Tarzetta sepultarioides*	LY:NV 2017-08-03 Type	MN712289	MN712244
*Tarzetta sepultarioides*	LY:NV 2017-08-16	MW862284	-----
*Tarzetta urceolata*	HKAS 127116	OQ422966	OQ418060
*Tarzetta victoriana*	M. Sánchez 2115 ITCV	PP825408	PP825451
*Tarzetta victoriana*	M. Sánchez 1752 Type ITCV	PP825409	PP825452
*Tarzetta* sp.	LY:NV 2018-06-12	MN712305	MN712263
*Tarzetta* sp.	RH1646	MN712326	MN712286

## Data Availability

All newly generated sequences were deposited in GenBank (https://www.ncbi.nlm.nih.gov/genbank/ (accessed on 25 January 2024)).

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
