# Peer review of "Eleven New Species of the Genus Tarzetta (Tarzettaceae, Pezizales) from Mexico"

_jof, 2024, doi:10.3390/jof10060403_

Round 1

Reviewer 1 Report

This article has reached the level that could be published.

This article about Tarzetta from Mexico is very detailed and gives new insights of the morphological, ecological, distribution information of this genus. I like this article very much. In the supplemental version, we provide annotations of the original text, and some small modification points are shown in the text.

Please add GenBank Accesion number before officially published.

Author Response

Thank you very much for all the support, as well as the observations made in this document so that it can be published and make known the news that we present.

For research article

Response to Reviewer X Comments

1. Summary

Thank you very much for taking the time to review this manuscript.

2. Questions for General Evaluation

Reviewer’s Evaluation

Response and Revisions

Does the introduction provide sufficient background and include all relevant references?

Yes

Are all the cited references relevant to the research?

Yes

Is the research design appropriate?

Yes

Are the methods adequately described?

Yes

Are the results clearly presented?

Yes

Are the conclusions supported by the results?

Yes

3. Point-by-point response to Comments and Suggestions for Authors

Comments 1: This article has reached the level that could be published.

Response 1: Thank you very much for your observations.

Comments 2: This article about Tarzetta from Mexico is very detailed and gives new insights of the morphological, ecological, distribution information of this genus. I like this article very much. In the supplemental version, we provide annotations of the original text, and some small modification points are shown in the text.

Response 2: Thank you very much for all your comments and observations made in the document. Appropriate modifications have already been made to the text, as he points out, and the GenBank accession numbers of all samples have also been added to the document.

Comments 3: Please add GenBank Accesion number before officially published.

Response 3: GenBank accession numbers have already been added to both the table and the text.

Reviewer 2 Report

This points out the high species diversity in this genus. The paper is well illustrated.

Please consider that the diagnoses should give information that separates the species under consideration from others. It is fine to give spore ranges etc. but keep in mind that you are trying to help the user of your work. In some cases the notes are more like a diagnosis. Some reorganization here.

Terminology needs to be consistent.

There are several issues outline below.

Commentary on “Eleven new specie3ss of the genus Tarzetta….

The language throughout needs to be reviewed carefully for word order and correct terminology. It is beyond a scientific reviewer to be able to correct the manuscript for English usage and proper grammar.

Line 35 and following: Somewhere in the introduction and elsewhere in the paper by Kumar et al. 2017 should be mentioned. This is on the Tarzetta – Geopyxis lineage and is found in This is in Fungal Biology 121: 264-284. This paper provides a placement of Tarzetta, sister to a group consisting of several taxa of the Pyronemataceae. It also provides some of the specific evidence that the species of Tarzetta are mycorrhizal since in there analysis several sequences from roots have been included.

Line 56: The authors fail to mention that there are species described from Asia – T. linzhiensis, T. tibetensis, T. urceolata. Are these excluded for some reason or were they merely overlooked? It is also worth noting that neither T. brasiliensis nor T. microspora have been verified. They are present in Indexfungorum but that is not a verification.

Figure 1 may not be clear. Editors should check this.

Line 168. These diagnoses should all be checked. They follow neither sentence structure nor descriptive structure. Note that apothecium is misspelled in this line. Also the use of rugulate seem odd. The normal usage might be rugose. I would suggest the following wording for this diagnosis.

“Differs from Tarzetta cupularis by smaller, sessile apothecia, crenate margin, narrower ellipsoid to oblong ascospores that are slightly rugose in SEM, growing under Quercus sp.”

I see that the section under notes is much more diagnosic and might be used in place of the diagnosis given which is not very specific when compared with what is said in the notes section. This brief diagnosis separates this species from T. cupularis but it does not distinguish from other species. This critique is applicable to all the descriptions. Editors should take note.

Line 176 – but occurs not but lives.

Line 180 and throughout the manuscript – “Medullar” is used here but elsewhere the more appropriate Medullary is used. Please correct.

Line181 and throughout the manuscript – “texture intricate” is used. This should be “textura intricata” perhaps this is autocorrect but it needs to be changed throughout. Elsewhere one finds “texture angulare” this should be “textura angularis.”

Line 187 and throughout the manuscript – Habit is listed as ectomycorrhizal. Although that may well be the case you do not have evidence in this specific case. One would assume since other members of the genus are mycorrhizal that this species is as well but it should not be stated here in habit or rather habitat. This should be a comment on were and with what plants the fungus was collected. One notes that under T. cupressicola there is not indication that it is mycorrhizal. One presumes that this reflects that Cupressus is not an ectomycorrhizal host. Perhaps there should be some comment in the notes on this observation.

Line 197 and following – I suppose these note might be helpful but one suspects that the key would be the important means of identification. Yes, they mostly distinguish among the several species that might be confused. The syntax becomes difficult.

Line 264 – I do not understand the terminology used here. “cracks until a hole is formed at the subsrate base.” The apothecium splits?  Please describe better.

Line 339 and elsewhere – The standard tissue type terminology has been attempted in most of the desciptions but here the term “texture pseudoparenchymatous” is used. I suppose that is textura angularis but is there some other significants?

Line 359 – This is not a diagnosis one gets no sense what is special about this taxon. Again, these diagnoses need to be reconsidered.

Line 450 – “cracked toward the center” please explain.

Line 685,686 – rather than ectal excipule it would probably be better to say “receptacle”

A word about the SEM photos. The surface ornamentation seems to be very fine. Describing this as rugose or rugulose may be overstating the observation. Certainly it would be appropriate to qualify the term, very finely rugose.

Author Response

Thank you very much for all the support, as well as the observations made in this document so that it can be published and make known the news that we present.

3. Point-by-point response to Comments and Suggestions for Authors

Comments 1: Please consider that the diagnoses should give information that separates the species under consideration from others. It is fine to give spore ranges etc. but keep in mind that you are trying to help the user of your work. In some cases the notes are more like a diagnosis. Some reorganization here.

Response 1: We agree with this comment. Therefore, the pertinent adjustments were made to all diagnoses, as recommended.

Comments 2: Terminology needs to be consistent.

Response 2: Agree. The terminology was reviewed and adjusted so that it was uniform throughout the document.

Comments 3: The language throughout needs to be reviewed carefully for word order and correct terminology. It is beyond a scientific reviewer to be able to correct the manuscript for English usage and proper grammar.

Response 3: A review was carried out on the grammar of the document, as well as its terminology, so that it is clear for the reader.

Comments 4: Line 35 and following: Somewhere in the introduction and elsewhere in the paper by Kumar et al. 2017 should be mentioned. This is on the Tarzetta – Geopyxis lineage and is found in This is in Fungal Biology 121: 264-284. This paper provides a placement of Tarzetta, sister to a group consisting of several taxa of the Pyronemataceae. It also provides some of the specific evidence that the species of Tarzetta are mycorrhizal since in there analysis several sequences from roots have been included.

Response 4: Added reference to Kumar et al. 2017, mention is also made of the position of the gender according to this document.

Comments 5: Line 56: The authors fail to mention that there are species described from Asia – T. linzhiensis, T. tibetensis, T. urceolata. Are these excluded for some reason or were they merely overlooked? It is also worth noting that neither T. brasiliensis nor T. microspora have been verified. They are present in Indexfungorum but that is not a verification.

Response 5: Regarding the indicated species, they were already included in the document, as well as in the phylogeny, in addition to adding the corresponding references. With respect to South American species, clarification is made in the conclusion section that a study is needed for verification.

Comments 6: Figure 1 may not be clear. Editors should check this

Response 6: Figure 1 was modified to see it better and make it more understandable.

Comments 7: Line 168. These diagnoses should all be checked. They follow neither sentence structure nor descriptive structure. Note that apothecium is misspelled in this line. Also the use of rugulate seem odd. The normal usage might be rugose. I would suggest the following wording for this diagnosis.

“Differs from Tarzetta cupularis by smaller, sessile apothecia, crenate margin, narrower ellipsoid to oblong ascospores that are slightly rugose in SEM, growing under Quercus sp.”

I see that the section under notes is much more diagnosic and might be used in place of the diagnosis given which is not very specific when compared with what is said in the notes section. This brief diagnosis separates this species from T. cupularis but it does not distinguish from other species. This critique is applicable to all the descriptions. Editors should take note.

Response 7: Modifications were made to the diagnoses of all species, to denote the main characters that distinguish the species. Thus, the terminology indicated to us was also corrected and standardized throughout the document.

Comments 8: Line 176 – but occurs not but lives.

Response 8: We agree, it was modified as suggested.

Comments 8: Line 180 and throughout the manuscript – “Medullar” is used here but elsewhere the more appropriate Medullary is used. Please correct.

Response 8: We agree, it fits "Medullary" throughout the document as it suggests.

Comments 9: Line181 and throughout the manuscript – “texture intricate” is used. This should be “textura intricata” perhaps this is autocorrect but it needs to be changed throughout. Elsewhere one finds “texture angulare” this should be “textura angularis.”

Response 9: We agree, it was modified to "textura intrincata" and "textura angularis" throughout the document, whether that is the case, as you suggest to us.

Comments 10: Line 187 and throughout the manuscript – Habit is listed as ectomycorrhizal. Although that may well be the case you do not have evidence in this specific case. One would assume since other members of the genus are mycorrhizal that this species is as well but it should not be stated here in habit or rather habitat. This should be a comment on were and with what plants the fungus was collected. One notes that under T. cupressicola there is not indication that it is mycorrhizal. One presumes that this reflects that Cupressus is not an ectomycorrhizal host. Perhaps there should be some comment in the notes on this observation.

Response 10: It was omitted to mention that they are mycorrhizal in the species, which, as mentioned, we do not have enough evidence to ensure this in all species. In the case of T. cupressicola, we also mention that we do not ensure that it has mycorrhiza, in the notes as in the discussion.

Comments 11: Line 197 and following – I suppose these note might be helpful but one suspects that the key would be the important means of identification. Yes, they mostly distinguish among the several species that might be confused. The syntax becomes difficult.

Response 11: It was taken into account and appropriate modifications were made.

Comments 12: Line 264 – I do not understand the terminology used here. “cracks until a hole is formed at the subsrate base.” The apothecium splits?  Please describe better.

Response 12: The wording of that character in the species was modified so that it could be better understood.

Comments 13: Line 339 and elsewhere – The standard tissue type terminology has been attempted in most of the desciptions but here the term “texture pseudoparenchymatous” is used. I suppose that is textura angularis but is there some other significants?

Response 13: We agree and the terminology was modified, as suggested, throughout the document.

Comments 14: Line 359 – This is not a diagnosis one gets no sense what is special about this taxon. Again, these diagnoses need to be reconsidered.

Response 14: This diagnosis was modified, as in all species for better understanding, according to what was suggested.

Comments 15: Line 450 – “cracked toward the center” please explain.

Response 15: The phrase was modified for better understanding.

Comments 16: Line 685,686 – rather than ectal excipule it would probably be better to say “receptacle”

Response 16: We agree and the term was modified to the most appropriate.

Comments 17: A word about the SEM photos. The surface ornamentation seems to be very fine. Describing this as rugose or rugulose may be overstating the observation. Certainly it would be appropriate to qualify the term, very finely rugose.

Response 17: We agree, the term was modified in all species, as suggested to us.

4. Response to Comments on the Quality of English Language

Point 1:

Response 1: A review of the English language in the document was made and we believe that the writing is now good.

Round 2

Reviewer 2 Report

The manuscript seems to be in good order.

No comment